# PsAF5 functions as an essential adapter for PsPHB2-mediated mitophagy under ROS stress in *Phytophthora sojae*

Wenhao Li [1,3], Hongwei Zhu [1,3], Jinzhu Chen[1,3], Binglu Ru[1], Qin Peng[1], Jianqiang Miao [1] ✉ & Xili Liu [1,2] ✉

Host-derived reactive oxygen species (ROS) are an important defense means to protect against pathogens. Although mitochondria are the main intracellular targets of ROS, how pathogens regulate mitochondrial physiology in response to oxidative stress remains elusive. Prohibitin 2 (PHB2) is an inner mitochondrial membrane (IMM) protein, recognized as a mitophagy receptor in animals and fungi. Here, we find that an ANK and FYVE domain-containing protein PsAF5, is an adapter of PsPHB2, interacting with PsATG8 under ROS stress. Unlike animal PHB2 that can recruit ATG8 directly to mitochondria, PsPHB2 in *Phytophthora sojae* cannot recruit PsATG8 to stressed mitochondria without PsAF5. *PsAF5* deletion impairs mitophagy under ROS stress and increases the pathogen's sensitivity to $H_2O_2$, resulting in the attenuation of *P. sojae* virulence. This discovery of a PsPHB2-PsATG8 adapter (PsAF5) in plant-pathogenic oomycetes reveals that mitophagy induction by IMM proteins is conserved in eukaryotes, but with differences in the details of ATG8 recruitment.

Reactive oxygen species (ROS) produced by the plant innate immune system in response to pathogen infection is an important means of disease resistance[1]. Plant cell-surface pattern recognition receptors (PRRs) recognize pathogen-associated molecular patterns (PAMPs) and apoplastic effectors and activate the downstream *Respiratory Burst Oxidase Homologue D* (*RBOHD*) encoded NADPH oxidase[2,3]. This enzyme catalyzes the transfer of electrons to molecular oxygen ($O_2$), generating superoxide anions ($O_2^-$), which are subsequently converted to hydrogen peroxide ($H_2O_2$) by superoxide dismutase (SOD)[4].

Host-derived ROS can promote protein and phenolic cross-linking and callose deposition in plant cell walls, thus limiting the infection by the pathogen. Among ROS, the most abundant, stable, and neutral molecule[5], $H_2O_2$, can also spread to pathogen cells through aquaporins to damage macromolecules and organelles[6]. Mitochondria, as centers of cellular biosynthetic and energy metabolism, are not only a major intracellular source of ROS, but also can be a target of ROS damage[7].

ROS can result in mitochondrial DNA (mtDNA) damage, impairment of the electron transport chain (ETC), release of cytochrome c (Cytc), and activation of mitochondrial caspase-mediated apoptosis signaling[7]. Damaged mitochondria can further exacerbate ROS release, necessitating their timely removal by pathogens and mitigation of associated side effects during host infection[8,9]. As the primary energy source for eukaryotes, maintaining mitochondrial homeostasis is crucial for cell survival.

Mitophagy, a special form of autophagy, is an important way for cells to selectively eliminate damaged mitochondria[10,11]. ROS stimulate the biogenesis of autophagosomes and acidic lysosomes/vacuoles[12], facilitating the renewal of damaged mitochondrial proteins and the removal of mitochondrial debris via mitophagy[13,14]. Mitophagy was shown to be important for pathogenesis and ROS stress response in the plant pathogenic fungus *Magnaporthe oryzae*[15,16]. Mitophagy includes two types, one of which is ubiquitin-dependent, while the

[1]State Key Laboratory for Crop Stress Resistance and High-Efficiency Production, College of Plant Protection, Northwest A&F University, Yangling 712100 Shaanxi, China. [2]Department of Plant Pathology, College of Plant Protection, China Agricultural University, 2 Yuanmingyuanxi Road, Beijing 100193, China. [3]These authors contributed equally: Wenhao Li, Hongwei Zhu, Jinzhu Chen. ✉e-mail: mjq2018@nwafu.edu.cn; seedling@nwafu.edu.cn

second is a ubiquitin-independent mechanism termed receptor-mediated mitophagy.

The most well-studied model of the ubiquitin-dependent process is the PINK1-PRKN/Parkin pathway in animals[17]. Upon mitochondrial damage, PINK1 protein degradation is inhibited, leading to its accumulation on the outer mitochondrial membrane (OMM)[18]. E3 ubiquitin ligase PRKN/Parkin is subsequently recruited to the OMM, where it ubiquitinates OMM proteins, allowing for the recruitment of ATG8s (LC3B) to mediate mitophagy through ubiquitin-binding receptor proteins[19]. SQSTM1/p62 (sequestosome 1), NBR1, CALCOCO2/NDP52 (calcium binding and coiled-coil domain 2), TAX1BP1 (Tax1 binding protein 1), and OPTN (optineurin) are currently reported to be ubiquitin-binding receptors[18]. The recruited ATG8 plays a central role in assembly of the phagophore (autophagosome precursor), in cargo recruitment, and in autophagosome membrane bilayer maturation through its lipidation with phosphatidylethanolamine (PE) or phosphatidylserine (PS)[20]. The translocation of ATG8 and the transition between its unlipidated and lipidated forms serve as a key indicator of autophagy-related processes[21,22].

However, while ubiquitin-dependent PINK1-PRKN/ Parkin-mediated mitophagy is conserved in animals, this mechanism has not been identified in plants or fungi. In contrast, Ubiquitin-independent mitophagy appears to be more conserved across species. This second major mechanism is also known as receptor-mediated mitophagy[17] and primarily involves the recruitment of ATG8-related proteins by OMM proteins through the ATG8 interaction motif (AIM). It has been found in fungi that ATG32, ATG43 and ATG24 OMM receptors can directly recruit ATG8 through the AIM motif to activate mitophagy[15,23], which is essential for maintaining the virulence and oxidative stress responses of fungal pathogens[24]. The inner mitochondrial membrane (IMM) protein PHB2 found in the plant-pathogenic fungus *Colletotrichum higginsianum* can interact with ATG24 and participate in mitophagy, but without a direct interaction with ATG8[23]. Interestingly, PHB2 has also been reported to be an IMM mitophagy receptor in animals, where it could recruit ATG8 directly to damaged mitochondria. But its direct interaction with ATG8 required SUMO modification, and there is no evidence of SUMO or ubiquitination modification in vivo[25]. Because IMM proteins have less commonly been found as a mitophagy receptors than OMM proteins, the question of whether PHB2 can be used as a mitophagy receptor in different species and whether an adapter protein is needed to act as a bridge in the process of ATG8 recruitment has not been well studied.

*Phytophthora sojae* is a devastating plant pathogenic oomycete responsible for causing widespread soybean blight[26]. *P. sojae* and its host *Glycine max* (soybean) have been widely used as a model in the study of pathogen-host interactions[27,28]. During infection, ROS produced by the host as a defense response poses a challenge for *P. sojae*, threatening its survival and plant colonization[29–31]. It has been extensively studied how pathogen effector proteins suppress host-derived ROS production[32]. However, questions remain about how plant pathogenic oomycetes maintain ROS tolerance and mitochondrial homeostasis under oxidative stress during infection[9,33].

Mitochondrial homeostasis is achieved mostly by mitophagy. However, *P. sojae* lacks critical classical components of the ubiquitin-dependent mitophagy apparatus, including PINK1, PRKN/Parkin, and ATG24[34]. Therefore, it is not clear how PsATG8 is recruited to mitochondria during the process of mitophagy in response to oxidative stress in *P. sojae*. In mammals, the FYVE and ANK domain-containing protein ANKFY1 (rabankyrin-5) is believed to affect mitochondrial homeostasis[35,36].

In this study, we identified and characterized PsAF5 which is an FYVE and ANK domain-containing protein. PsAF5 is crucial for the virulence of *P. sojae*, and *PsAF5* knockout mutants displayed increased sensitivity to ROS. Using Immunoprecipitation-Mass Spectrometry (IP-MS) methods, we identified PsAF5 as an adapter protein of IMM

mitophagy receptor PHB2, serving as a bridge for PHB2 to recruit ATG8. This work highlights IMM mitophagy as an important role in the process of pathogen-host interactions, and reveals PHB2 to be a key IMM receptor for mitophagy in oomycetes. Although PHB2 exhibits some functional similarity with orthologs in other eukaryotes as an IMM mitophagy receptor, the identification of an adaptor protein for PHB2 recruitment of ATG8 makes this IMM receptor-mediated autophagy pathway more diverse from both regulatory and evolutionary perspectives.

## Results

### Expansion of *AF* genes in oomycetes
ANK and FYVE domain-containing proteins (AFs) are conserved in four eukaryotic taxonomic groups[37]. Manual annotation revealed that representative oomycete species from the *Phytophthora*, *Peronospora*, and *Pythium* genera typically possess 10-12 AFs[38]. *Nitzschia inconspicua* (diatoms) and *Ectocarpus siliculosus* (brown algae), which belong to the same kingdom, Stramenopila, as oomycetes[39], have 4 and 2 AFs, respectively. However, only one AF is present in representative species of metazoan, viridiplantae, and fungal (Supplementary Fig. 1a). *AF* gene evolution is relatively conserved in different species of oomycetes, *P. sojae* has 10 AF proteins, named PsAF1-10 based on the conservation of their FYVE domains (Supplementary Fig. 1b). Genes *PsAF4-8* cluster together with their oomycete orthologs, and have a distant affinity with *AF* genes in fungi and plants (Supplementary Fig. 1c), while *AF1-3, 9-10* form an oomycete-specific clade.

PsAFs can be classified into two groups based on domain combinations. The first group, containing PsAF1-4, 7-10, similar to plants, contains only ANK and FYVE domains. The second group (PsAF5 and PsAF6) has a third domain, resembling animal and fungal AFs. However, the combination with a tetratricopeptide repeat (TPR) domain found in PsAF5 and PsAF6 is unique to oomycetes and *N. inconspicua* (Fig. 1a). A quantitative polymerase chain reaction (qPCR) assay revealed that the genes in one clade, *PsAF4*, *PsAF5*, *PsAF6*, *PsAF7*, and *PsAF8*, exhibited high transcript levels in zoospores and cysts or during cyst germination (Supplementary Fig. 1d; Supplementary Table 1), suggesting their possible importance in early infection processes.

### *PsAF5* is essential for the virulence of *P. sojae*
We used the CRISPR/Cas 9 method to generate homozygous knockout mutants of all individual *PsAF* genes (ΔPsAF1-10 mutants) and verified them using PCR amplification (Supplementary Fig. 2a, b; Supplementary Tables 2–4). The colony morphologies and in vitro growth rates of individual *PsAF* gene knockout mutants were similar to those of control transformants (CK) (Supplementary Fig. 2c, d). Similarly, in all ten *PsAF* gene knockouts, the morphology of sporangia and oospores were not significantly changed, and the zoospore release process was also not significantly different from that of the wild-type, P6497 (Supplementary Fig. 2e; Supplementary Table 5). On the other hand, soybean infection tests revealed that the virulence of the ΔPsAF1, ΔPsAF5, ΔPsAF7, ΔPsAF9, and ΔPsAF10 mutants were all reduced by more than 40%. Notably, the two *PsAF5* knockouts, ΔPsAF5-1 and ΔPsAF5-3, showed more than 80% loss of virulence (Fig. 1b; Supplementary Fig. 2f). Furthermore, ΔPsAF5, ΔPsAF7, and ΔPsAF9 mutants showed a significant correlated reduction in numbers of both sporangia and zoospores reduction compared to the wild-type P6497 (Supplementary Fig. 2g, h). Zoospores germination rates in ΔPsAF5 mutants was markedly reduced compared to the wild-type P6497 (Supplementary Fig. 2i). Interestingly, the ΔPsAF5 mutants exhibited a significant increase in oospore number compared to the wild-type P6497 (Supplementary Fig. 2j).

Due to the pronounced virulence phenotype of ΔPsAF5 mutants, we selected PsAF5 for further characterization. For complementation assays, we transformed full-length *PsAF5* into the knockout mutant ΔPsAF5-3 to generate a complementary transformant C-*PsAF5*

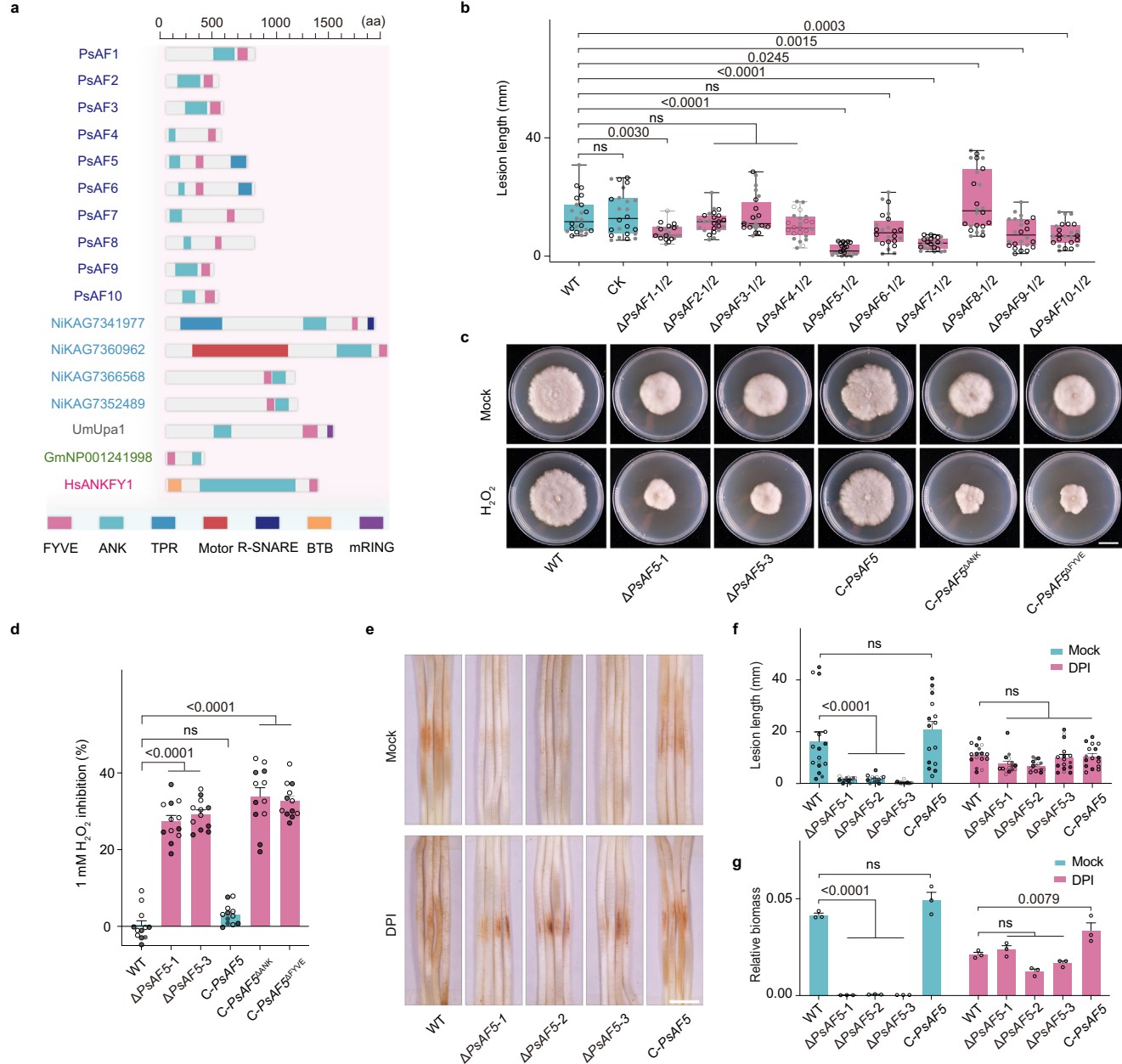

**Fig. 1 | ΔPsAF5 mutants are defective in H₂O₂ tolerance and plant infection.**
**a** Domain structures of AFs. AFs are shown from *Phytophthora sojae* (blue), *Nitzschia inconspicua* (light blue), *Ustilago maydis* (gray), *Glycine max* (green), and *Homo sapiens* (magenta). **b** Virulence of different *PsAF* knockout (Δ*PsAF*) mutants of *P. sojae*. Circles and grey dots indicate data from two biological replicates (n = 24 biologically independent samples). The line within the box in box plots represents the median value, while the bounds of the box indicate the 25th and 75th percentiles. Additionally, whiskers extend from these bounds to encompass the minimum and maximum values. **c** Sensitivity to 1 mM H₂O₂ of Δ*PsAF5* mutants and *PsAF5* complementary transformants; complementation was with full length *PsAF5* (C-*PsAF5*) or ANK or FYVE domain truncated *PsAF5* (C-*PsAF5*ᐃANK and C-*PsAF5*ᐃFYVE). Scale

bar, 1 cm. **d** Quantitation of inhibition by 1 mM H₂O₂. Circles and grey dots indicate two biological replicates (n = 12 biologically independent samples). **e–g** DPI (diphenyleneiodonium) treatment compensates for the virulence loss of Δ*PsAF5* mutants. **e** Infection lesions on etiolated soybean seedlings, 48 h post-inoculation. **f** Quantitation of virulence as indicated by lesion length from two biological replicates (n = 16 biologically independent samples). **g** Quantitation of virulence as indicated by genomic DNA qPCR measurements of biomass (n = 3 biologically independent samples). All experiments in (**b–g**) were independently repeated 3 times with similar results. Data are presented as mean value ± SEM. Ordinary one-way ANOVA and Dunnett's multiple comparisons tests were used, ns non-significant. Source data are provided as a Source Data file.

(Supplementary Fig. 3a, b). In this transformant, expression of full-length *PsAF5* fully rescued the defects observed in three independent Δ*PsAF5* mutants (Supplementary Fig. 2d–j; Supplementary Fig. 3c–j; Supplementary Table 6). Complementation with a *GFP-PsAF5* fusion (C-*GFP-PsAF5*) also rescued the defects of Δ*PsAF5-3*, but complementation with deletions of *PsAF5* lacking the ankyrin or FYVE domains (C-*PsAF5*ᐃANK and C-*PsAF5*ᐃFYVE, respectively) failed to rescue those defects (Supplementary Fig. 3c–j; Supplementary Table 6).

## Enhanced ROS sensitivity caused the reduced virulence of ΔPsAF5

To investigate the cellular pathways involving PsAF5, we subjected Δ*PsAF5* mutants to different stresses. The sensitivity to osmotic stress exerted by sorbitol was increased in the two Δ*PsAF5* mutants, which was rescued by the complementation with full-length *PsAF5* but not by the *PsAF5*ᐃANK or *PsAF5*ᐃFYVE alleles (Supplementary Fig. 4a, b). In both mycelial growth and spore germination assays, Δ*PsAF5* mutants also

had increased sensitivity to $H_2O_2$, with defects restored to wild-type levels by complementation with full-length *PsAF5* (Fig. 1c, d; Supplementary Fig. 4c).

During the early stages of infection, the host produces $H_2O_2$ at *P. sojae* infection sites in order to prevent pathogen infection[29–31]. The wild-type strains exhibited normal infection, while the mutant Δ*PsAF5*-3 was restricted to initial spore germination sites, and failed to successfully infect the host (Supplementary Fig. 4d). To confirm the possible relationship between the reduced virulence of the Δ*PsAF5* mutants and their increased sensitivity to $H_2O_2$, we conducted diphenyleneiodonium (DPI) inhibition tests. DPI selectively inhibits plasma membrane NADPH oxidase, which is essential for generating extracellular ROS in both plant and mammalian cells[33,40] (Supplementary Fig. 4d). DPI treatment significantly inhibited $H_2O_2$ accumulation around infection sites (Supplementary Fig. 4d) and substantially compensated for the virulence deficiency of Δ*PsAF5* mutants (Fig. 1e–g; Supplementary Table 7). These results suggested that the decreased virulence of Δ*PsAF5* mutants may be associated with their increased sensitivity to $H_2O_2$.

### Deletion of *PsAF5* inhibits pathogen autophagy induced by ROS

To further elucidate the functional mechanism underlying the response to ROS stress, we examined the subcellular localization of GFP-tagged PsAF5 under $H_2O_2$ treatment using complemented *P. sojae* strain C-*GFP-PsAF5*. Generally, PsAF5 exhibited cytoplasmic localization when cultured in V8 medium. However, after $H_2O_2$ treatment, GFP-tagged PsAF5 displayed punctate co-colocalization with BFP (blue fluorescent protein)-tagged PsATG8 but did not co-localize with mCherry-labeled early endosomes (EE), late endosomes (LE), or peroxisomes (Supplementary Fig. 5a).

A rapid increase in the levels of lipidated ATG8 (ATG8-PE) compared to ATG8, a marker of autophagy activity, demonstrated that ROS could indeed increase the autophagy levels of *P. sojae*, with the autophagy level peaking about 5 min after $H_2O_2$ treatment in the wild-type, P6497 (Fig. 2a, b). After deletion of *PsAF5*, autophagy increased more slowly after $H_2O_2$ treatment than in the wild-type (Fig. 2a, b). This suggested that PsAF5 may participate in the autophagy pathway, needed for the ROS resistance of *P. sojae*.

### PsAF5 interacts with ATG8 via its AIM1 motif

To investigate the potential direct interaction between PsAF5 and PsATG8, we performed co-immunoprecipitation (Co-IP) and pull-down assays. Our Co-IP experiments indicated a weak interaction between PsAF5 and PsATG8 in vivo, which could be enhanced upon $H_2O_2$ treatment (Fig. 2c). Furthermore, we successfully purified both proteins using an *Escherichia coli* expressed system, and demonstrated their direct interaction in vitro (Fig. 2d).

Tandem-tag methods, such as mCherry-GFP fusions, are frequently employed to monitor autophagy[41]; the GFP signal is quenched in the acidic lumen of lysosomes and vacuoles, while the mCherry fluorescence remains stable[41]. In this study, we employed this approach by connecting a tandem mCherry-GFP reporter to PsAF5 to track its localization during autophagy (Fig. 2e). It is important to note that cells maintain a baseline level of autophagy. Therefore, we observed that PsAF5 could formed some condensates where the GFP signal was quenched in the acidic lumen, while the mCherry fluorescence remained stable. These condensates represent autophagosome structures that could be labeled by BFP-ATG8 (Supplementary Fig. 5b). We also observed punctate co-localization between GFP-labeled PsAF5 and BFP-PsATG8 in response to $H_2O_2$ treatment or during plant infection. This suggests the formation of newly generated autophagic structures that have not undergone acidification (Fig. 2f), providing further evidence for the interaction between PsAF5 and PsATG8 under these conditions. However, due to the expected quenching of GFP at low pH, GFP labeling was not intended to detect PsAF5 within acidified

autophagosomes. Therefore, for the subsequent studies in this article, we primarily utilized mCherry labeling.

Since AIM is the primary motif that mediates protein interactions with ATG8, we also used the tandem mCherry-GFP reporter to identify which AIM motif(s) or domain(s) is responsible for the localization of PsAF5 during autophagy (Fig. 2e). By mutating the two key amino acids at positions 3 and 6 in each of four possible AIM motifs (MuAIM1-4), we identified AIM1 as the key functional AIM motif in PsAF5. In *MuAIM1* mutants, but not *MuAIM2-4* mutants, we observed a significant decrease in the co-localization PsAF5 and PsATG8 localization in acidified compartments (Fig. 2g, h), and detected the loss of interaction between PsAF5 and PsATG8 in vivo and in vitro conditions (Fig. 2i, j). The sensitivity of the *MuAIM1* mutants to $H_2O_2$ was significantly increased compared with that of the wild-type P6497, while no significant differences were observed in the other three AIM mutants (Supplementary Fig. 6a). AIM1-mutated *PsAF5* was unable to complement the reduced virulence caused by the deletion of *PsAF5* (Supplementary Fig. 6b), consistent with the $H_2O_2$ sensitivity results. These findings suggested that PsAF5 is mainly localized to autophagosomes through AIM1 interactions with ATG8, and that PsAF5 mainly exerts its effects on ROS responses and virulence via its involvement in autophagosome structures.

### ROS enhanced the distribution of PsAF5 and PsATG8 in mitochondria

To further investigate the role of PsAF5 in autophagy, we linked PsAF5 to a Flag-tag and identified candidate interacting proteins through Immunoprecipitation-Mass Spectrometry (IP-MS). Interestingly, a large number of mitochondrial proteins were among the candidate interacting proteins (Supplementary Table 8), suggesting a functional link between PsAF5 and mitochondria.

Considering that oxidative stress can activate mitophagy, we examined the subcellular localization of PsAF5 and PsATG8 relative to mitochondria. When *P. sojae* was cultured in V8 medium, PsAF5, PsATG8, and mitochondria did not show colocalization. However, upon treatment with $H_2O_2$ or the mitochondrial complex III inhibitor ametoctradin, which induces endogenous ROS production, PsAF5 and PsATG8 colocalized with mitochondria. This was accompanied by a shift in mitochondrial morphology from linear to punctate structures (Fig. 3a).

The localization of PsAF5 to mitochondria following $H_2O_2$ treatment was further confirmed by labeling mitochondria with mitochondrial proteins PsCytc (cytochrome c) and PsCoadh (glutaryl-CoA dehydrogenase) (Supplementary Fig. 5c). Additionally, Mito-tracker labeling of mitochondria during *P. sojae* infection of soybean revealed an increase in the localization of PsAF5 and PsATG8 to mitochondria (Fig. 3b). Furthermore, other factors known to induce ROS production, such as aging and nitrogen-starvation, also resulted in increased localization of PsAF5 to mitochondria (Fig. 3c). Immunoblotting results demonstrated the stability of the tandem protein without loss of the fluorescence tag (Supplementary Fig. 7).

To examine the mitochondrial distribution of PsAF5 at the protein level, active and intact mitochondria were isolated using sucrose extraction centrifugation (Fig. 3d, e, f). The IMM protein PsPHB2 and cytoplasmic protein ACTB (β-Actin) were used as indicators of successful mitochondrial extraction (Fig. 3g). Immunoblotting assays revealed an increase in mitochondrial distribution and a decrease in cytoplasmic distribution of PsAF5 following $H_2O_2$ treatment (Fig. 3g, h). Fluorescence localization observations and isolated mitochondrial protein assays collectively demonstrated that PsAF5 exhibits increased localization to mitochondria under ROS stress.

### Deletion of *PsAF5* leads to loss of ROS-induced mitophagy

We evaluated whether PsAF5 was required for ROS-induced mitophagy in *P. sojae*. The Δ*PsAF5* mutant showed a loss of ROS-induced

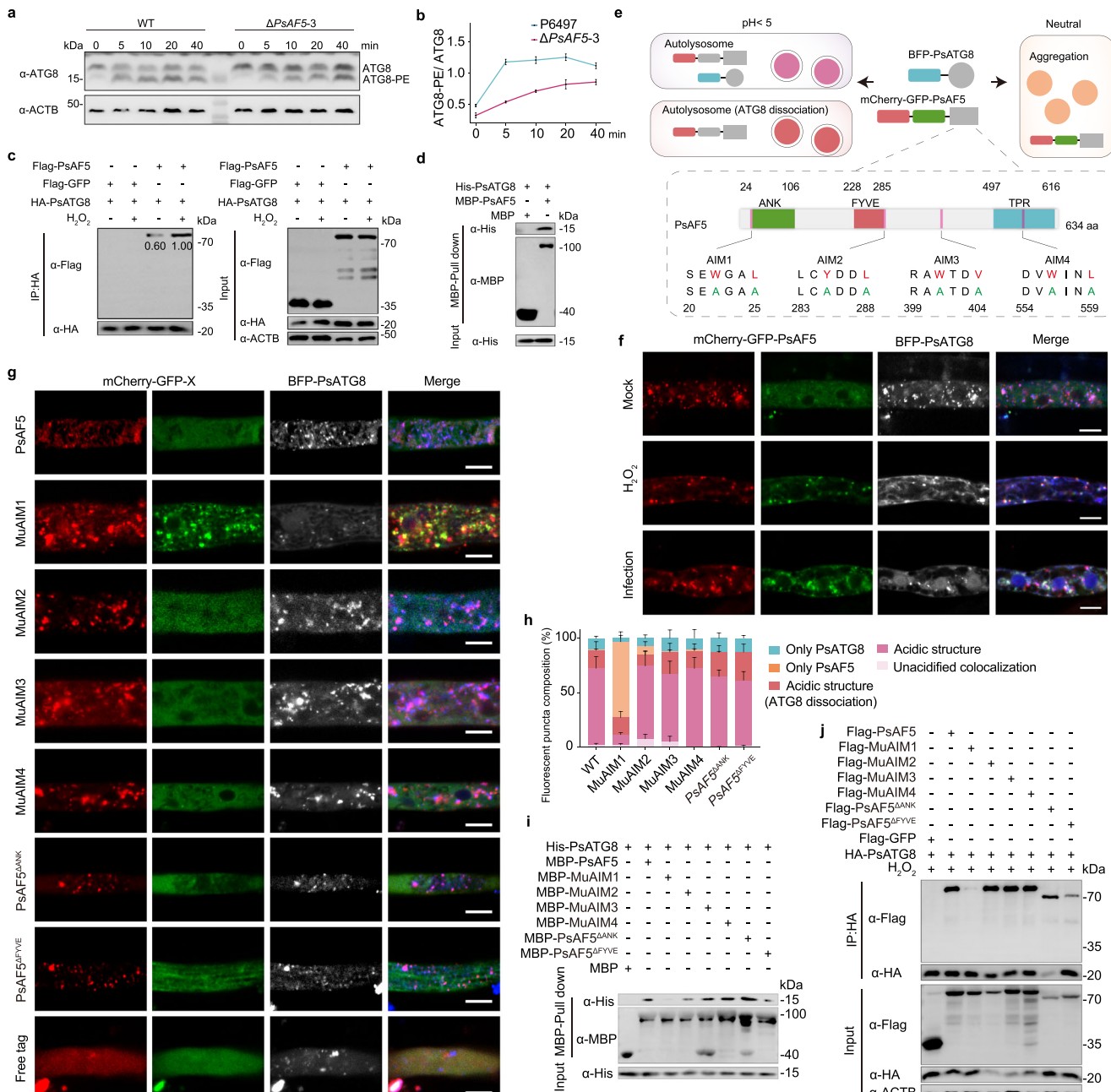

**Fig. 2 | PsAF5 interacts with and co-locates with PsATG8 through an AIM1 motif. a** *PsAF5* deletion reduces $H_2O_2$-induced intracellular ATG8 activation. The proteins detected by the ATG8 antibody were isolated using urea SDS–PAGE. **b** The ratio of ATG8-PE to ATG8 after 1 mM $H_2O_2$ treatment over time; *n* = 3 biologically independent samples. **c** Interaction of PsAF5 and PsATG8 in vivo detected by a Co-IP assay. $H_2O_2$ treatment was 1 mM for 1 h. **d** Interaction of PsAF5 and PsATG8 in vitro detected by a pull-down assay. **e** Schematic diagram illustrating PsAF5 tagged in tandem with mCherry and acid-sensitive GFP while ATG8 is tagged on the N-terminus with a BFP (blue fluorescent protein) tag. The enlarged area shows the four AIM motifs of PsAF5 and their respective mutation sites. **f** PsAF5 and ATG8 colocalize into puncta during infection (2 h post inoculation) or $H_2O_2$

treatment (1 mM 1 h). **g** Co-localization of BFP-PsATG8 with different PsAF5 domain and AIM motif mutant proteins, observed by fluorescence microscopy, without any $H_2O_2$ or infection treatment. Scale bar, 5 μm (**f**, **g**). **h** The distribution of fluorescence colors in the experiments shown in (**g**), as described in the Methods and illustrated in Fig. 2e. Experiments in (**a**, **c**, **d**, **g**, **f**) were independently repeated 3 times with similar results. **i** Interaction of PsATG8 with different PsAF5 domain and AIM motif mutant proteins in vitro, detected by a pull-down assay. **j** Interaction of PsATG8 with different PsAF5 domain and AIM motif mutant proteins in vivo, detected by a Co-IP assay. $H_2O_2$ treatment was 1 mM for 1 h. Experiments in (**i**, **j**) were independently repeated twice with similar results. In (**b**) and (**h**), error bars in graphs represent means ± SEM. Source data are provided as a Source Data file.

mitophagy, as measured by immunoblotting analysis of the mitochondrial matrix proteins, Apoptosis-Inducing Factor B (AIF B) and Cytc, as well as the IMM protein, Mitochondrial Carrier (MC) (Fig. 4a–c). The deficiency of Δ*PsAF5* mutants in ROS-induced mitophagy was also detected by monitoring the ratio of mitochondrial DNA to nuclear DNA (Fig. 4d, e). The reduction in numbers of mitochondria in the wild-type P6497 also was prevented by the vacuolar acidification

inhibitor bafilomycin A1 (Baf A1) (Fig. 4f, g). Furthermore, we found that the mitochondrial complex III inhibitors pyraclostrobin and ametoctradin also caused a decrease in the number of mitochondria in the wild-type strains, which was prevented by the deletion of *PsAF5* (Fig. 4h, i). The sensitivity of the hyphal growth of the three Δ*PsAF5* mutants to these two agents was also significantly increased compared to the wild-type P6497 (Fig. 4j, k).

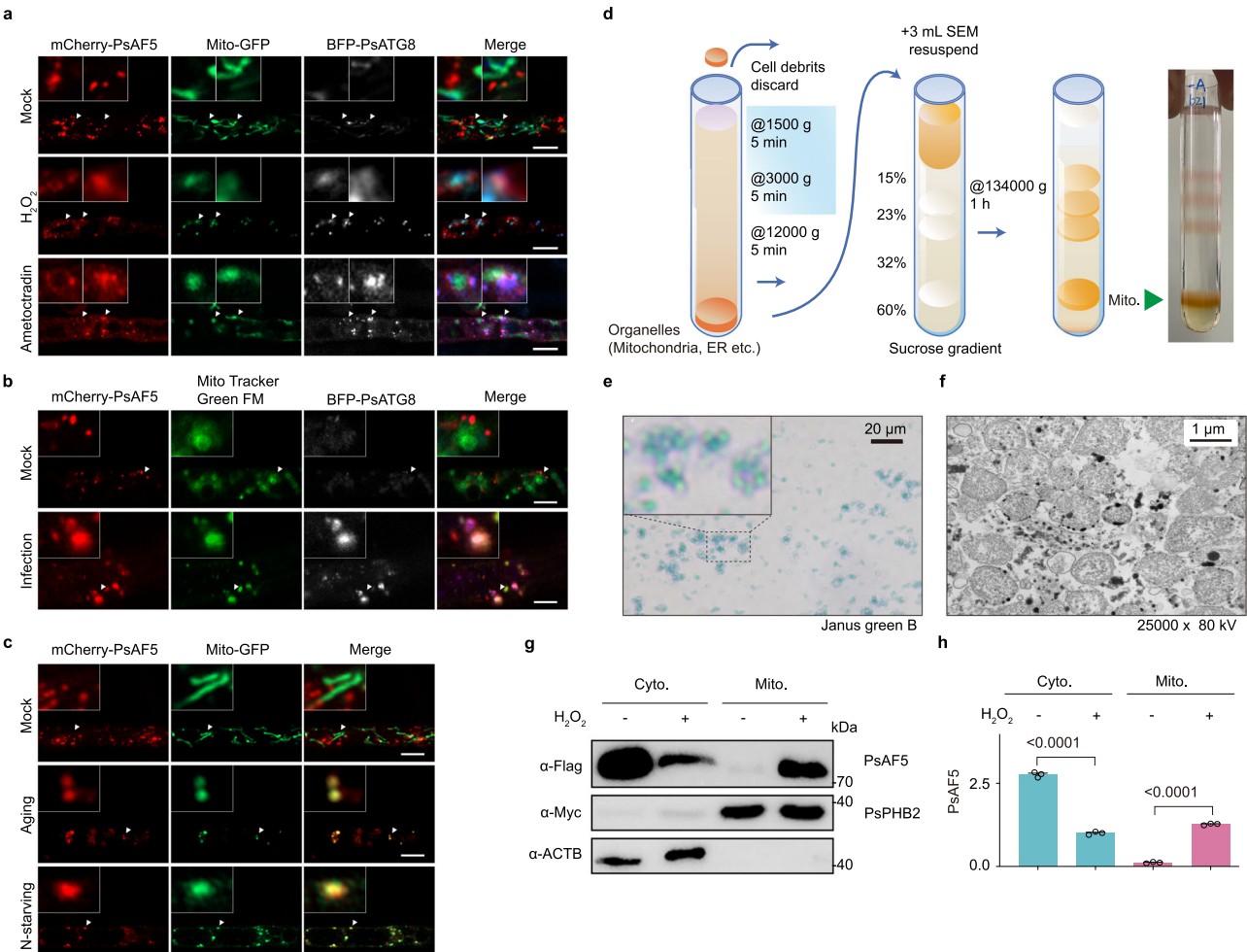

**Fig. 3 | Localization of PsAF5 and PsATG8 to mitochondria under different ROS-inducting conditions. a** Both $H_2O_2$ treatment and mitophagy-inducting conditions can increase the mitochondrial localization of PsAF5 and PsATG8. The treatment time for 1 mM $H_2O_2$ or 4 μg/mL ametoctradin was 1 h prior to observation. **b** The mitochondrial localization of PsAF5 and PsATG8 increased during infection. Prior to observation, the mycelia were treated with Mito Tracker Green FM for 1 h. **c** Mitophagy inducting conditions, aging, and N-starvation, can increase the mitochondrial localization of PsAF5. In (**a**–**d**), scale bars represent 5 μm. **d** Schematic diagram of the extraction of mitochondria from *P. sojae* using sucrose gradient centrifugation. **e** Verification of mitochondrial activity. The isolated mitochondria in (**d**) method were stained with Janus green B. **f** Mitochondrial integrity verification. The mitochondria isolated in (**d**) method were observed by transmission electron microscopy (TEM). **g, h** $H_2O_2$ treatment can promote the transfer of PsAF5 to mitochondria from the cytoplasm. Error bars in graphs represent means ± SEM (*n* = 3 biologically independent samples). Data in (**a**, **b**, **c**, **d**, **g**, **h**) represent at least 3 biological experiments, and data in (**e**, **f**) represent 2 biological experiments. Ordinary one-way ANOVA and Dunnett's multiple comparisons test were used. Source data are provided as a Source Data file.

To explore the role of PsAF5 in mitophagy, we generated a mitochondrial targeting signal fused with a tandem mCherry-GFP tag (Mito-mCherry-GFP) to visualize mitochondria within acidic compartments[42,43]. Mitochondria in acidic autophagosomes appear as red-only structures due to GFP bleaching (Fig. 5a). First, as a control, we produced a *P. sojae ATG7* knockout mutant (Supplementary Fig. 8a), and confirmed that the *ATG7* mutant could not activate autophagy levels before or after $H_2O_2$ treatment (Fig. 5b). Using the *ATG7* knockout transformant as the background material to express the Mito-mCherry-GFP mitophagy indicator, we observed that the *ATG7* knockout transformant displayed linear and granular mitochondrial morphology when cultured on V8 medium, and exhibited yellow fluorescence (overlapping green and red) indicating lack of acidification. Upon treatment with $H_2O_2$, mitochondria in the Δ*ATG7* mutant became round without GFP acidification and bleaching, and without recruitment of BFP-PsATG8, yielding yellow fluorescence (Fig. 5c). In contrast, in the wild-type P6497 under $H_2O_2$ stress, mitochondria displayed stress-induced acidification and rounding, with recruitment of BFP-PsATG8 to stressed mitochondria, resulting in purple fluorescence (overlapping red and blue). However, the addition of autophagy inhibitor bafilomycin A1 prevented the bleaching of GFP in mitochondria and the recruitment of BFP-PsATG8 (Fig. 5d), thus showing a phenotype of mitophagy retardation consistent with that of Δ*ATG7* mutant under $H_2O_2$ stress. These results supported that Mito-mCherry-GFP could correctly report mitophagy in *P. sojae*.

Applying this method of expressing tandem fluorescent tags, we observed that both the wild-type and the Δ*PsAF5* mutants displayed linear and granular mitochondrial morphology when cultured on V8 medium, consistent with previous studies[43], and exhibited yellow fluorescence consistent with lack of mitophagy. Upon treatment with $H_2O_2$ or pyraclostrobin, the mitochondria of the wild-type P6497 displayed stress-induced acidification and rounding, with recruitment of BFP-PsATG8 to stressed mitochondria, resulting in purple fluorescence. In contrast, mitochondria in the Δ*PsAF5*-3 mutant became round without GFP acidification and bleaching, and the recruitment of BFP-PsATG8 was absent, yielding yellow fluorescence. This defect in the Δ*PsAF5*-3 was rescued by the expression of *Flag-PsAF5* (Fig. 5e, f).

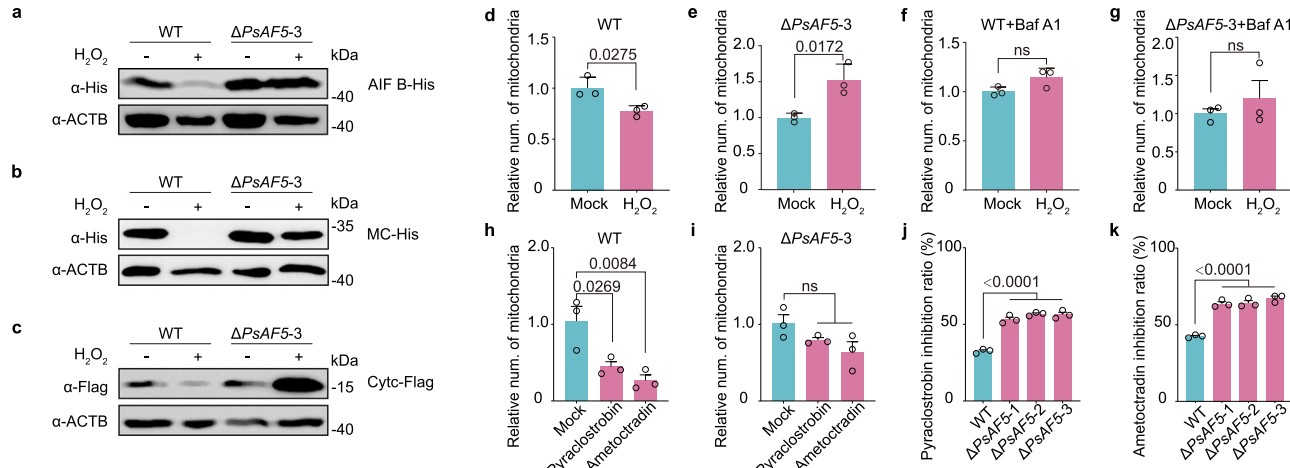

**Fig. 4 | ROS-induced mitophagy is blocked in the ΔPsAF5 mutant.**
**a–c** Immunoblot analysis of mitophagy markers in the wild-type P6497 and the ΔPsAF5 mutant mycelium under H₂O₂ treatment. The analysis of total proteins used His-tag antibodies against AIF B (a mitochondrial matrix protein) (**a**) and mitochondrial carrier (MC) (an inner mitochondrial membrane protein) (**b**), and using anti-Flag antibodies against Cytc (a mitochondrial matrix protein) (**c**). The tagged proteins were expressed in P6497 and ΔPsAF5 by ectopic over-expression. ACTB antibody was used as a loading control. A representative example of three replicates is shown for (**a–c**). **d–g** The relative number of mitochondria was quantified by using qPCR to determine the ratio of mitochondrial DNA to nuclear DNA in each strain, which were subjected to treatment with 0 or 1 mM H₂O₂ for 36 h with or without 20 nM bafilomycin A1 (an autophagy inhibitor). Two-tailed Student's $t$ test

was used ($n = 3$ biologically independent samples, mean ± SEM; ns non-significant). **h, i** The relative number of mitochondria was measured by the ratio of mitochondrial DNA to nuclear DNA in the indicated strains treated with pyraclostrobin (2 μg/mL) or ametoctradin (4 μg/mL), $n = 3$ biologically independent samples. **j, k** The sensitivities of ΔPsAF5 mutants to mitophagy inducers were significantly increased. The effect of 2 mg/L pyraclostrobin or 4 mg/L ametoctradin on hyphal growth of the wild-type P6497 and ΔPsAF5 mutants were assessed after 7 d post-inoculation ($n = 3$ biologically independent samples). Experiments in (**d–k**) were independently repeated twice with similar results. Bar plots with mean ± SEM, ordinary one-way ANOVA and Dunnett's multiple comparisons test were used; ns non-significant. Source data are provided as a Source Data file.

## PsAF5 is an adapter of IMM protein PsPHB2

One of the PsAF5-interacting proteins identified by IP-MS analysis was PsPHB2 (Supplementary Table 8), which has been implicated in mitophagy in animals and pathogenic fungi[18,23]. The interaction between PsAF5 and PsPHB2 was confirmed through pull-down (Fig. 6a) and Co-IP (Fig. 6a) assays, and it was found that H₂O₂ treatment enhanced this interaction in vivo (Fig. 6b). Moreover, the interaction between PsAF5 and PsPHB2 under H₂O₂ treatment was inhibited upon addition of epoxomicin, an OMM rupture inhibitor (Fig. 6c).

To further investigate the interaction between PsAF5 and PsPHB2, we performed pull-down experiments with different fragments of PsAF5, which showed that the ANK domain mediates the interaction between PsAF5 and PsPHB2 (Fig. 6d). In order to verify the interaction region of PsAF5 with PsPHB2 on PsAF5 in vivo, we co-expressed Flag-tagged fragments of PsAF5 with Myc-tagged PsPHB2 in *P. sojae*. Then immunoprecipitation (IP) was performed using Flag-trap and Myc-trap beads. The results confirmed that the ANK domain was the key region of PsAF5 interacting with PsPHB2 (Fig. 6e, f), which was consistent with the pull-down results. Furthermore, the results of mitochondrial extraction and immunoblotting showed that deletion of the ANK domain decreased the recruitment of PsAF5 into mitochondria under ROS treatment (Fig. 6g). Further phenotypic analysis showed that expression of PsAF5^ΔANK failed to compensate for the reduced virulence and zoospore numbers nor increased ROS sensitivity and oospore numbers of the ΔPsAF5 mutant (Fig. 1c, d; Supplementary Fig. 3, 4b, c; Supplementary Table 6).

## PsAF5 as a bridge for PsPHB2 to recruit PsATG8

To test whether PHB2 functioned as a mitophagy receptor to recruit ATG8 in *P. sojae* as in animals[25], we conducted Co-IP tests in vivo. Complexes involving PsPHB2 and PsATG8 could not detected in the untreated wild type, though complexes involving activated (lipidated) PsATG8 could be weakly detected upon H₂O₂ treatment. Notably, no complexes involving PsPHB2 and PsATG8 could be detected in the

ΔPsAF5-3 mutant, regardless of whether or not it had been treated with H₂O₂ (Fig. 7a). Pull-down assays revealed that the level of His-PsATG8 pulled down with PsPHB2-MBP depended on the amount of PsAF5-His present (Fig. 7b), suggesting that PsAF5 could promote the interaction between PsPHB2 and PsATG8 in a dose-dependent manner. In the resting state, a certain amount of ATG8 was localized to mitochondria (Fig. 7c, d). When different *PsAF5* alleles were expressed in the ΔPsAF5-3 strain, only alleles with an intact AIM1 motif and ANK domain could increase the mitochondrial localization of PsATG8 and PsAF5 (Fig. 7e). We used the activation ratio of lipidated to non-lipidated ATG8 (ATG8-PE/ATG8) in purified mitochondria as an indicator of the level of mitophagy. In the wild-type P6497, the level of mitochondrial PsATG8 activation increased significantly upon H₂O₂ addition, whereas the ΔPsAF5-3 mutant exhibited a marked decrease in mitochondrial PsATG8 activation. Cytoplasmic PsATG8 activation following H₂O₂ treatment was unchanged by *PsAF5* deletion (Fig. 7c, d). We used the strong HAM34 promoter to obtain a transformant overexpressing *PsAF5*, in which the transcript level of *PsAF5* was 8 times higher than that in the wild-type P6497 (Supplementary Fig. 8b; Supplementary Table 9). The levels of mitophagy in this transformants were significantly increased relative to the wild-type P6497 under H₂O₂ stress, as evidenced by increased lipidation of ATG8 (Supplementary Fig. 8c, d). We further examined the interaction between PsPHB2 and activated (lipidated) PsATG8 and observed that higher levels of activated (lipidated) PsATG8 bound to PHB2 could be detected in the PsAF5 overexpression transformant than in wild-type P6497 (Supplementary Fig. 8e).

During infection, mitochondria exhibited acidification and PsATG8 recruitment in the the wild-type P6497, whereas the ΔPsAF5-3 mitochondria displayed changes in morphology associated stress but did not display acidification or BFP-PsATG8 recruitment (Fig. 7f, g). Mitophagy is associated with increased levels of mitochondrial protein ubiquitination. In the wild-type, H₂O₂ treatment led to a more than two-fold increase in mitochondrial protein ubiquitination levels

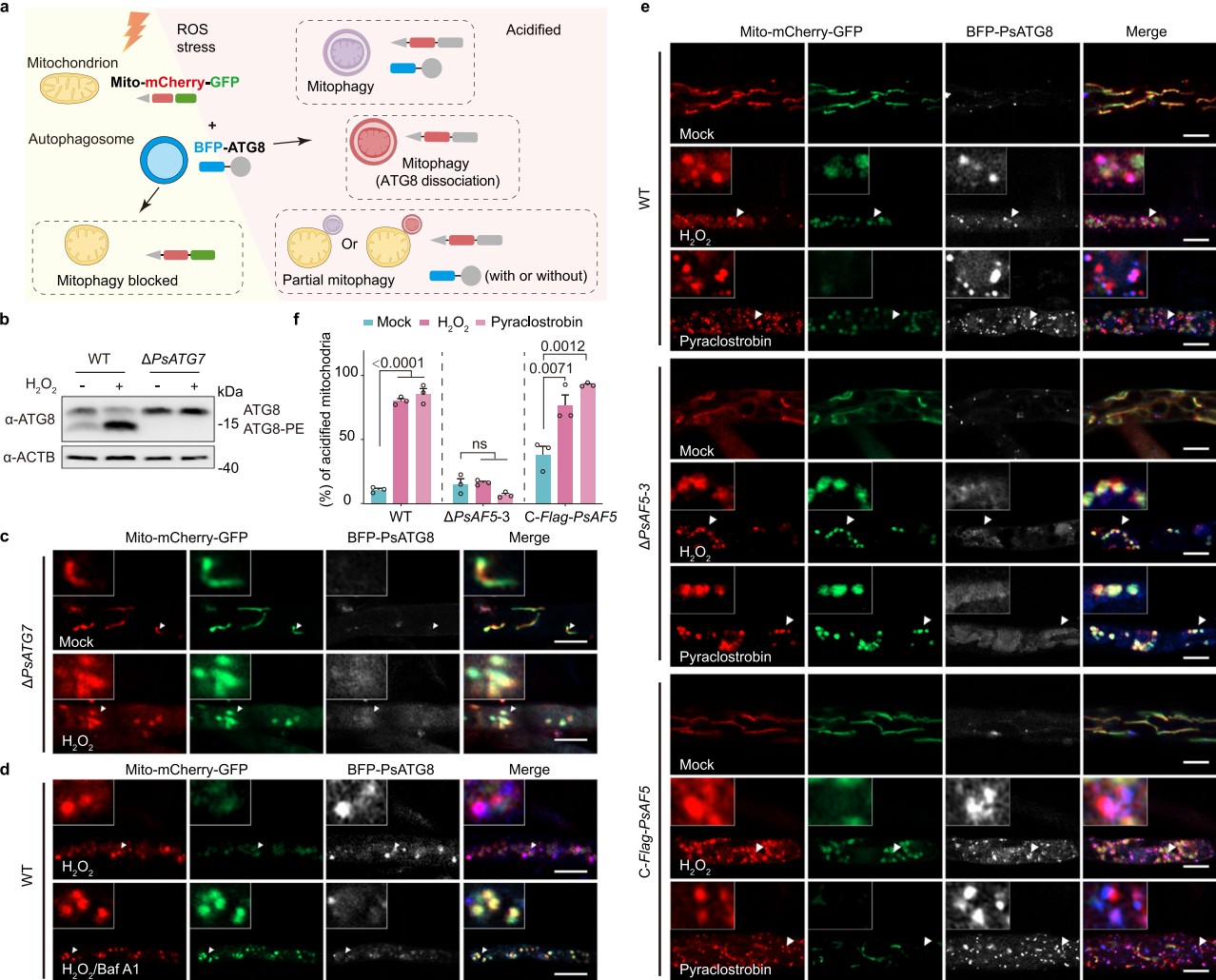

**Fig. 5 | ROS-induced mitochondrial acidification is blocked in the ΔPsAF5 mutant. a** The schematic diagram depicts possible stages of mitophagy. BFP-ATG8 was co-expressed in *P. sojae* strains together with a mitochondrial-targeting peptide fused with tandem acid-sensitive GFP and acid-insensitive mCherry. **b** *PsATG7* deletion can reduce the lipidation of ATG8 before and after $H_2O_2$ treatment (1 mM, 1 h). The proteins detected by the ATG8 antibody were isolated using urea SDS−PAGE. **c** BFP-PsATG8 and Mito-mCherry-GFP described in (**a**) were co-expressed and observed in a ΔPsATG7 mutant. The transformants were cultivated in V8 liquid medium or V8 supplied with 1 mM $H_2O_2$ for 1 h. **d** BFP-PsATG8 and Mito-mCherry-GFP described in (**a**) were co-expressed and observed in the wild-type

strain. The transformants were cultivated in V8 supplied with 1 mM $H_2O_2$ or V8 supplied with 1 mM $H_2O_2$ and 20 nM bafilomycin A1 for 1 h. **e** BFP-PsATG8 and Mito-mCherry-GFP described in (**a**) were co-expressed and observed in the wild-type, ΔPsAF5 mutants and PsAF5 complemented transformant. The transformants were cultivated in V8 liquid medium or V8 supplied with 1 mM $H_2O_2$ or 0.1 mM pyraclostrobin for 1 h. The scale bars are all 5 μm. **f** Percentage of mitochondrial acidification in (**e**). Data in (**b**−**f**) represent at least 3 biological experiments. Ordinary one-way ANOVA and Dunnett's multiple comparisons test were used; and ns non-significant. Data are presented as mean value ± SEM. Source data are provided as a Source Data file.

(Fig. 7h). In contrast, the ΔPsAF5-3 mutant showed less than a 40% increase, which did not even reach the levels in untreated wild-type (Fig. 7h).

## Discussion

Pathogens utilize various strategies to counteract host-derived ROS during infection. These strategies include the secretion of effectors that suppress ROS production, the secretion of antioxidant enzymes such as catalase for ROS degradation, and the enhancement of cellular tolerance to ROS stress[32]. Here, from a family of ankyrin-FYVE proteins that has expanded in oomycete species, we have identified a player in the regulation of mitophagy in response to ROS stress. This player, PsAF5 in the pathogenic oomycete *P. sojae*, acts as an adaptor for recruitment of ATG8 by the prohibin PHB2, forming a mitophagy regulatory axis, PsPHB2-PsAF5-PsATG8. This pathway facilitates the efficient removal of mitochondria damaged by oxidative stress, thereby ensuring the pathogen's normal physiological functions and

successful infection. This finding reveals important mechanistic details of how oomycete pathogens strengthen cellular ROS tolerance during infection.

Receptor-mediated mitophagy may involve a crucial class of proteins called the "adapters". Receptors and adapters share a similar evolutionary history. But unlike receptors, adapters do not degrade during autophagy[44]. Some adapter proteins have the ability to interact with ATG8 and guide targeted cargo to the autophagy pathway. For example, the human FYCO1 protein (FYVE and coiled-coil domain-containing protein) can interact with LC3B/ATG8 via an AIM motif[21,45]. We found that the amount of PsAF5 in mitochondria increased during autophagy and PsAF5 could interact with ATG8 through its AIM1 motif, recruiting ATG8 to damaged mitochondria. Furthermore, ATG8 was also almost undetectable on the mitochondria of the full-length *PsAF5* deletion or MuAIM1 mutants. No interaction between PsPHB2 and PsATG8 could be detected in ΔPsAF5 mutants. Thus, we define PsAF5 as a mitophagy adapter, which can amplify mitophagy signals and

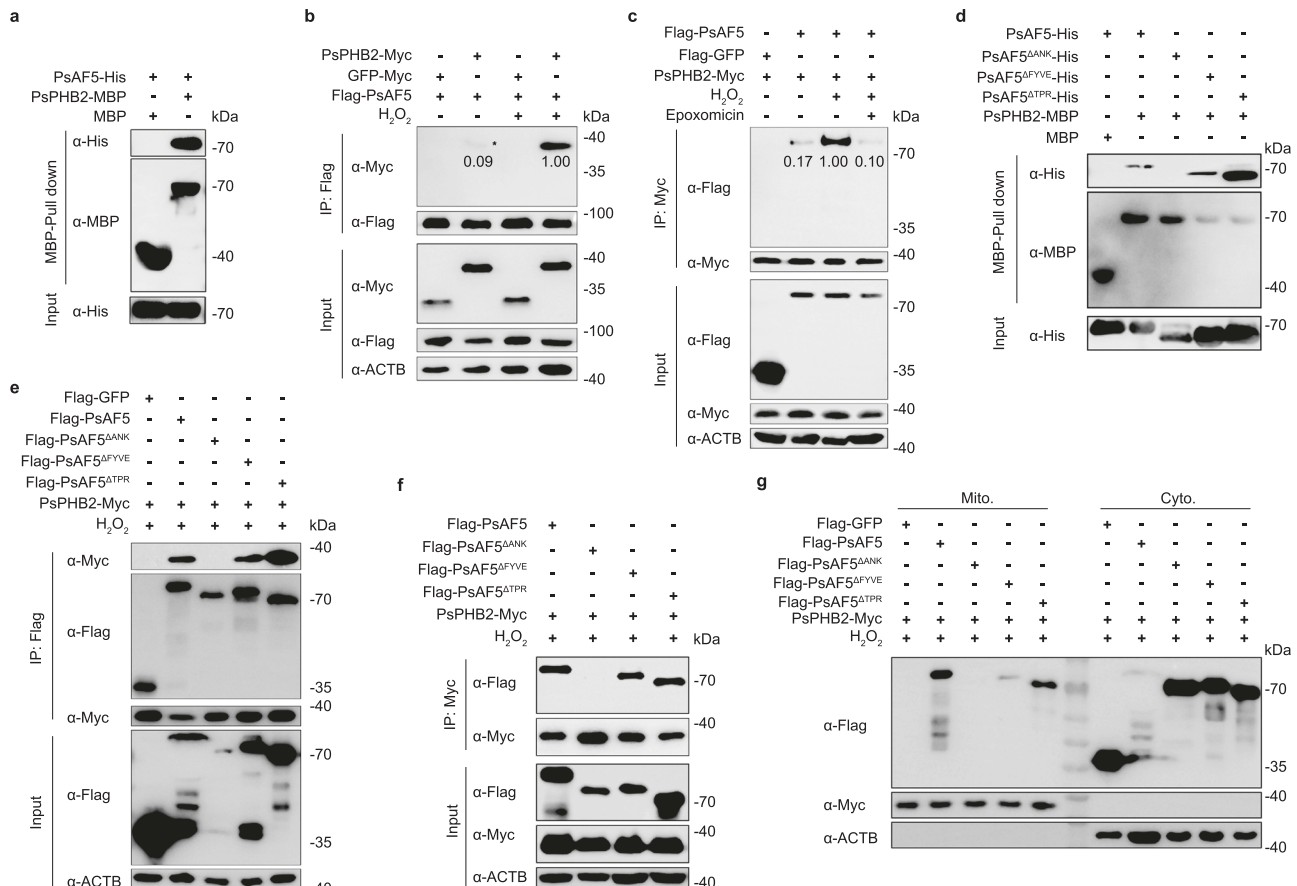

**Fig. 6 | PsAF5 is recruited to the mitochondria through its ANK domain interaction with PsPHB2. a** PsAF5 can interact with inner mitochondrial membrane (IMM) PsPHB2 in vitro. Pull-down assay using dextrin beads incubated with MBP-PsPHB2 and His-tagged PsAF5, followed by immunoblotting with anti-His and anti-MBP antibodies. **b** $H_2O_2$ treatment can increase the interaction between PsAF5 and PsPHB2. PsPHB2-Myc or GFP-Myc was co-expressed with Flag-PsAF5 in the wild-type P6497, with or without 1 h treatment with 1 mM $H_2O_2$, then the interaction was detected by immunoprecipitation with anti-Flag beads and subsequent immunoblotting with anti-Myc and anti-Flag antibodies. The numbers represent relative gray values of the immunoblot bands. **c** Inhibition of outer mitochondrial membrane (OMM) rupture can inhibit the interaction between PsAF5 and PsPHB2. The interaction was detected by immunoprecipitation with anti-Myc beads, and

subsequent immunoblotting with anti-Myc and anti-Flag antibodies. The interaction was further examined by treating the transformants with 1 mM $H_2O_2$, or 1 mM $H_2O_2$ plus 100 nM epoxomicin for 1 h. The numbers represent relative gray values. **d** The ANK domain in PsAF5 mediates the interaction of PsPHB2 in vitro. Pull-down assay using dextrin beads incubated with MBP-PsPHB2 plus His-tagged PsAF5 and its mutants, followed by immunoblotting with anti-His and anti-MBP antibodies. **e, f** The ANK domain in PsAF5 mediates the interaction of PsPHB2 in vivo. Mycelium was treated with 1 mM $H_2O_2$ for 1 h before co-immunoprecipitation with anti-Flag (**e**) or anti-Myc (**f**) antibodies. **g** The ANK domain is critical for PsAF5 to relocate to mitochondria. Mycelia were treated with 1 mM $H_2O_2$ for 1 h before mitochondrial isolation. The above experiments were independently repeated twice with similar results. Source data are provided as a Source Data file.

promotes PsATG8 recruitment to ROS-stressed mitochondria in oomycete pathogens (Fig. 8).

ROS stress induces mitochondrial dysfunction, depolarization, and OMM rupture, resulting in the exposure of IMM proteins[46]. Moreover, ROS stress has been reported as exposing the mitochondrial IMM protein PHB2, which can function as a receptor to recruit lipidated ATG8/LC3 directly to promote mitophagy in animals[17,25]. In the fungal pathogen *Colletotrichum higginsianum*, ChATG24 is recruited to mitochondria during mitophagy by prohibitins ChPHB1 and ChPHB2, although a direct interaction between ChATG8 and ChPHB2 was not detected[23]. In our study, we showed that IMM PsPHB2 acts as a mitophagy receptor protein in oomycetes in addition to animals and fungi. Additionally, we discovered that the protein PsAF5 serves as an adapter, bridging the recruitment of PsATG8 by PsPHB2. These findings suggest that the mechanism of PsPHB2-mediated recruitment of PsATG8 in oomycetes may differ from that in animals. Due to the instability of mitochondrial proteins during mitophagy, including PsPHB2, there is no suitable internal control for PsATG8 protein quantification in isolated mitochondria. Therefore, we employed the ratio of lipidated to unlipidated PsATG8 as an indicator of mitophagy

and demonstrated that PsAF5 is critical to the recruitment and activation of PsATG8 in mitochondria.

AF proteins are conserved across diverse taxonomic groups[37], suggesting that their function as an adapter protein of PHB2 may be widespread. Previous research has indicated that the animal AF protein, also known as ANKFY1/ Rabankyrin-5, impacts mitochondrial homeostasis[35,36]. ANKFY1/Rabankyrin-5 in animals may also function as mitophagy adapter as PsAF5, which might explain why PHB2 could interact only with purified ATG8 when expressed in animal cells treated with mitochondrial respiration inhibitors instead of being directly expressed in *E. coli* without SUMOylation[17,25].

The autophagosome is a bilayer membrane compartment that arises from de novo synthesis initiated at the pre-autophagosomal structure (PAS), a membraneless organelle with liquid-like properties[47]. A distinctive characteristic of autophagy is the high abundance of PI3P enrichment in autophagosome membranes[48]. Recent studies have revealed that mammalian FYVE protein ALFY is recruited to depolarized mitochondria through NIPSNAP1, a mitophagy receptor on mitochondria. Simultaneously, ALFY can interact with ATG8s through its FYVE domain and AIM motif, initiating

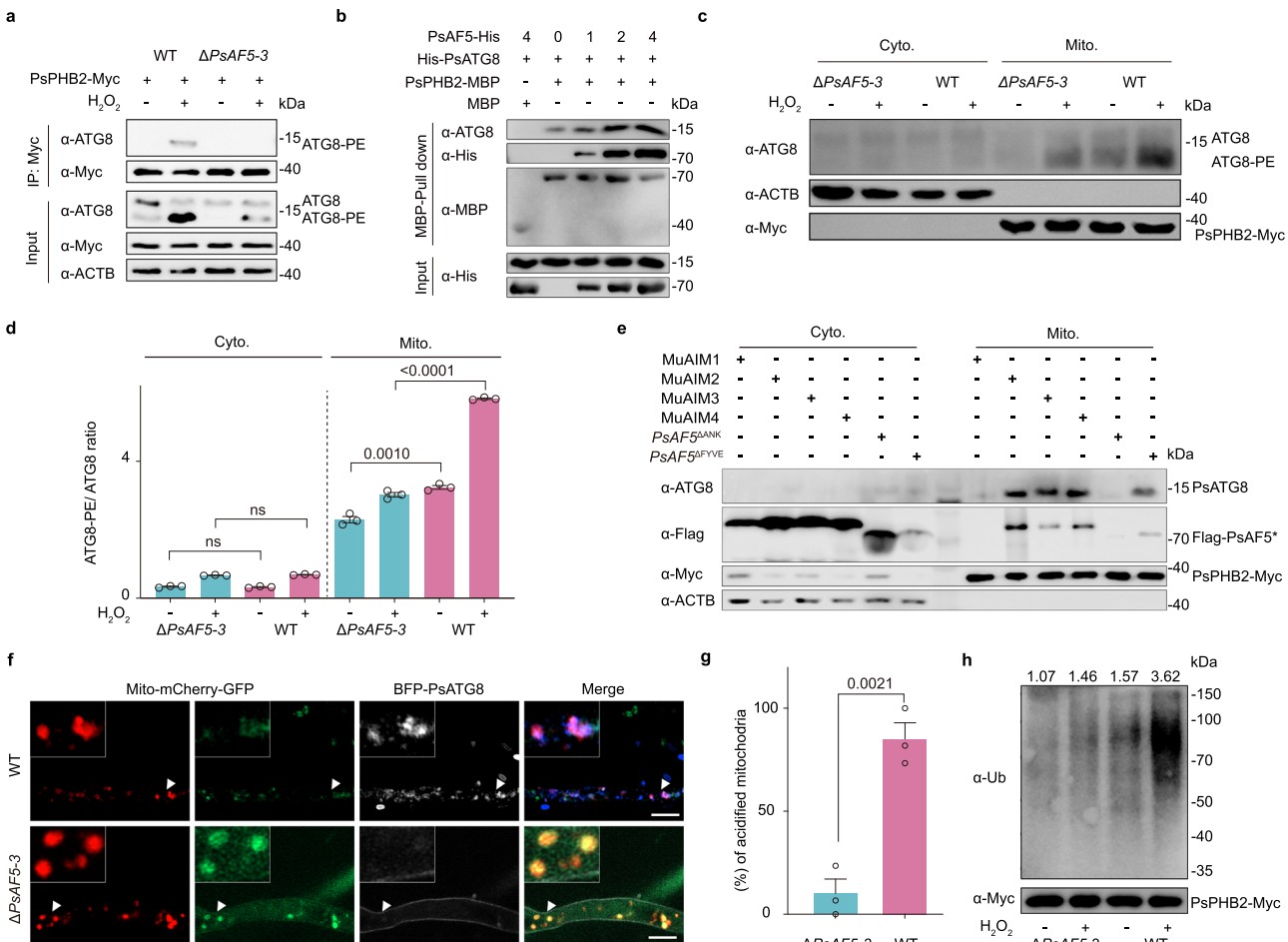

**Fig. 7 | PsAF5 serves as an adapter to aid recruitment of PsATG8 to stressed mitochondria. a** The interaction between PsPHB2 and PsATG8-PE in vivo requires PsAF5. 1 mM H₂O₂ was added 1 h before protein extraction in the treatment group. The proteins detected by the ATG8 antibody were isolated using urea SDS–PAGE. **b** PsAF5 assists the interaction of PsPHB2 and PsATG8 in vitro in a dose-dependent manner. Recombinant PsPHB2-MBP or MBP (control) immobilized on dextrin beads were incubated with His-PsATG8, and with increasing amounts of PsAF5-His (1×, 2×, 4×). Protein eluates from washed beads were used for immunoblotting with anti-MBP, anti-His or anti-ATG8 antibodies. **c, d** PsAF5 deletion can reduce the activation of ATG8 in mitochondria. The proteins detected by the ATG8 antibody were isolated using urea SDS–PAGE. The ratio of lipidated to non-lipidated PsATG8 (**d**) is based on the gray values of bands in (**c**) results (n = 3 biologically independent samples). Error bars in graphs represent means ± SEM. Two-tailed Student's *t* test

was used, and ns non-significant. **e** The distribution of ATG8 in mitochondria is decreased by either an AIM1 mutation or an ANK domain deletion in *PsAF5*. **f** The deletion of *PsAF5* resulted in the inhibition of mitophagy during infection; mitophagy is indicated by the colocation of ATG8 with acidified mitochondria. The scale bars are 5 µm. **g** The percentage of mitochondrial acidification in (**f**); results came from three independent experiments. Ordinary one-way ANOVA and Dunnett's multiple comparisons test were used, and ns non-significant. Data are presented as mean value ± SEM. **h** The deletion of *PsAF5* can reduce the level of mitochondrial ubiquitination (Ub). 1 mM H₂O₂ was added to mycelia 1 h before mitochondrial purification and protein extraction in the treatment group. The numbers represent relative gray values of ubiquitin-positive bands. Data in (**a**–**h**) represent at least 3 biological experiments. Source data are provided as a Source Data file.

mitophagy signaling[49]. In yeast, ATG18, ATG20, ATG21, and ATG24 can be recruited to the PAS by binding to PI3P[50]. In our study, we observed that PsAF5 colocalizes with PsATG8 and accumulates in acidified autophagosomal compartments in *P. sojae* mycelia and protoplasts. PsAF5 exhibits a distinct mode of interaction with PsATG8 requiring only the AIM1 motif, but not the participation of the FYVE domain. However, the FYVE domain of PsAF5 is essential for normal virulence, growth and H₂O₂ tolerance phenotypes, which suggests that this domain may perform a different function than binding PI3P and directing ATG8 to mitochondria. The localization of PsAF5 to autophagosomes under H₂O₂ conditions may represent the mechanism by which *P. sojae* detects ROS signals and prepares for potential mitochondrial damage resulting from subsequent oxidative stress. This process may involve post-translational modifications, such as phosphorylation. In summary, this research represents the mechanistic insights into mitophagy in plant pathogenic oomycetes in which the PsAF5 of *P. sojae* participates as an intermediate adapter.

Apart from its involvement in combating oxidative stress, PsPHB2 warrants further investigation regarding its potential roles in regulating the degradation of mitochondrial membrane proteins, assembling and stabilizing mitochondrial oxidative phosphorylation systems, maintaining the stability of the mitochondrial genome, and modulating the morphogenesis of mitochondrial cristae[51–54].

As a process involving an IMM protein, PHB2-mediated mitophagy in mammals requires the participation of PINK/Parkin ubiquitin ligase to label and degrade mitochondrial outer membrane proteins to achieve OMM rupture, which works through the protease system. However, previous studies have not detected ubiquitination of mammalian PHB2[25], and no ubiquitination was detected on PsAF5 and PsPHB2 in our study (data not shown). Nevertheless, the overall level of mitochondrial ubiquitination increased in the wild-type *P. sojae* strain upon H₂O₂ treatment, and this process was blocked in the Δ*PsAF5* mutants. This finding suggests that *P. sojae* may possess an E3 ubiquitin ligase that is distinct from PRKN/Parkin in animals yet

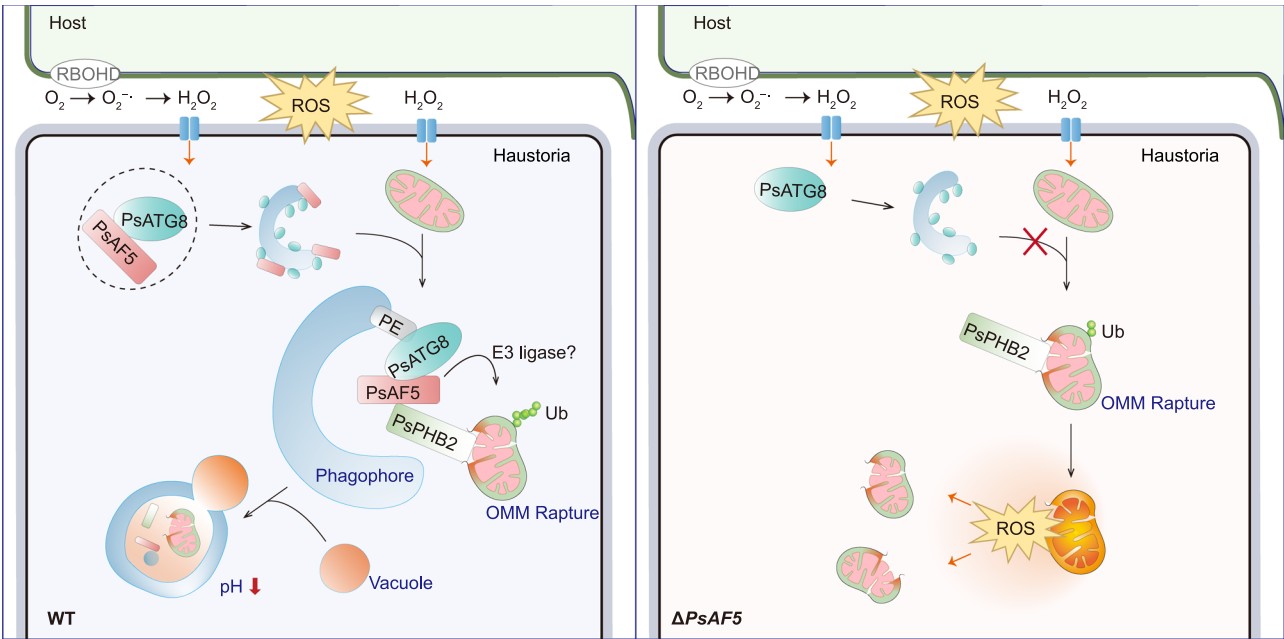

**Fig. 8 | Proposed model for the adapter role of PsAF5 in ROS-induced mitophagy.** After sensing infection by pathogens, host-derived ROS can be produced in the apoplastic space of plant cells. Of those ROS, $H_2O_2$ can be transported into the cytoplasm of haustoria through aquaporins on the plasma cell membrane of the pathogen. $H_2O_2$ elevation can promote PsAF5 interaction with PsATG8 through the AIM1 motif. ROS stress can also lead to mitochondrial rupture and the exposure of the IMM mitophagy receptor PsPHB2 to the cytoplasm, increasing the chance of PsAF5 contacting and interacting with PsPHB2 through its ANK domain. This interaction aids the recruitment of lipidated PsATG8 (PsATG8-PE) and the phagophore to stressed mitochondria, thereby promoting mitophagy. The autophagosome that envelops the mitochondrial further fuses with vacuoles, helping decrease the pH that activates acidic hydrolytic enzymes for the degradation of the stressed mitochondria. The Δ*PsAF5* mutant cannot efficiently remove damaged mitochondria through mitophagy in time, resulting in ROS accumulation and abnormal mycelial physiology. Ub indicates mitochondrial ubiquitination.

performs a similar function. Additionally, the rupture of the outer mitochondrial membrane involves E3 ubiquitin ligase-mediated proteasome-dependent degradation of the OMM[25]. It remains unclear whether PsAF5 can promote the ubiquitination process of OMM proteins as well as indirectly recruiting ATG8 to damaged mitochondria, thereby enhancing the intensity of mitophagy.

Oxidative stress from apoplastic $H_2O_2$ is one of the earliest host defenses encountered by *P. sojae* during soybean infection. Consistent with PsAF5 having a key protective role against oxidative stress, *PsAF5* transcription was highly elevated in the two principal infective stages of *P. sojae*, cysts and germinating cysts. In this work we simulated the environment of host-derived ROS during the infection stage of pathogens by direct exogenous addition of $H_2O_2$ or factors capable of inducing endogenous ROS production in *P. sojae*. Mitochondrial complexes I and III in the electron transport chain (ETC) are the primary sites for ROS production[55,56]. Inhibiting the activity of these complexes leads to mitochondrial ROS production and mitophagy[22]. Accordingly, mitochondrial complex III inhibitors ametoctradin and pyraclostrobin also promoted the mitochondrial recruitment of PsAF5 and PsATG8. Furthermore, Δ*PsAF5* mutants showed increased sensitivity to inhibitors of the mitochondrial respiratory chain compared to the wild-type P6497. Mitochondrial ROS production is also induced under conditions of nitrogen starvation and aging[57], and both conditions promoted the colocalization of PsAF5 with mitochondria. Thus the regulation of mitophagy by PsAF5 may involve regulatory mechanisms in response to both exogenous and endogenous ROS stress. Oospores have a thick-walled structure which can resistant to unfavorable environmental conditions[58]. Δ*PsAF5* mutants exhibited a significant increase in oospore production, suggesting potential crosstalk between $H_2O_2$ sensing mechanisms and the regulation of oospore production in *P. sojae*.

Mitochondria play crucial roles in energy and metabolism in eukaryotic organisms and are targeted by numerous commercial antimicrobial agents for control of eukaryotic pathogens and pests. Additionally, mitochondria are recognized as critical organelles for regulating cell survival and apoptosis. Mitochondrial depolarization can lead to the release of cytochrome c, which is a crucial component of the mitochondrial respiratory chain. This release can activate the apoptotic pathway[59]. Our findings in *P. sojae* reveal that the level of Cytc decreases due to the mitophagy as a result of $H_2O_2$ treatment in wild type, but there is a significant increase in Cytc in Δ*PsAF5* mutants, which is detrimental to pathogen survival. PsAF5 thus may play a role in regulating the balance between mitophagy and apoptosis, enhancing the ability of pathogens ability to cope with oxidative stress and promoting infection. This study's discovery of a unique mitochondrial stress response mechanism in oomycetes, which potentially differs from other eukaryotes such as animals and fungi, holds great potential for guiding the development of targeted oomycetes control agents.

## Methods

### Strains and culture conditions

The *P. sojae* P6497 strain was utilized as the wild-type in this study. All strains were cultured on 10% V8 vegetable medium in the dark at 25 °C[60]. Soybean (*Glycine max*) leaves and etiolated seedlings were obtained by sowing soybean seeds (Williams) and incubating them under 25 °C for 10 d with light or 4 d in darkness. The hypocotyl inoculation method was employed to assess the virulence of *P. sojae* strains and to observe the subcellular localization of fluorescent labeling during the infection stage.

### Plasmid construction

*PsAF5, PsPHB2, PsATG8, PsMC, PsCytc, PsAIF B, PsRab5, PsRab7*, and *SKL* genes were amplified from *P. sojae* cDNA and ligated into pYF3 (G418)

or pYF3OX (oxathiapiprolin) using In-Fusion Snap Assembly Master Mix (Takara) to express them as Flag, Myc, GFP, BFP, His, HA or mCherry fusion proteins. The amplified primer information is provided in Supplementary Table 10, and the tag sequences are provided in Supplementary Table 11. Mitochondrial targeting signal (MTS) sequences were amplified from *P. infestans β-ATPase* and ligated to pYF3 through enzymatic digestion to generate pYF3-MTS-mCherry, pYF3-MTS-GFP, and pYF3-MTS-mCherry-GFP constructs[43]. The *Flag-PsAF5^ΔANK* and *Flag-PsAF5^ΔFYVE* constructs were expressed in pYF3 and verified through PCR and western blot. Supplementary Table 12 lists all primers used for PsAF5 mutant vector construction.

### *P. sojae* transformation and phenotype analysis

Transformants were generated via PEG-CaCl$_2$ mediated protoplast transformation and screened using V8 medium containing 50 μg/mL G418 or 0.05 μg/mL oxathiapiprolin[61]. Genomic DNA was extracted from *P. sojae* mycelia using the CTAB protocol[62]. For mycelial growth, a 4.2 mm plug was placed on 10% V8 agar medium and incubated for 7 d at 25 °C in darkness. Oospore and sporangium production were quantified using a microscope at 200× magnification after culturing transformants on V8 plates for 7 d at 25 °C in darkness, with sporangium production determined after rinsing the plates eight times with sterile water. Zoospore production was measured by adding 5 mL sterile water to each plate after sporangium production and counting the number of zoospores per 1 μL using a microscope. All experiments were repeated at least three times.

### Total RNA extraction and qPCR analysis

Total RNA was extracted from the fresh transformed hyphae using the SV Total RNA Isolation kit (Promega) according to the manufacturer's instructions. The quality and quantity of the extracted RNA were determined using agarose gel electrophoresis and NanoDrop (Thermo), respectively. First-strand cDNA was synthesized from 1 μg of RNA using PrimeScript™ Reverse Transcriptase (Takara), and qPCR was performed using CFX Real-Time PCR (Bio-Rad) with TB Green Fast qPCR Mix (Takara). The 2$^{-ΔΔCT}$ method was employed to calculate the data[63], with the *actin* gene serving as the reference control based on previous studies[64]. Each qPCR assay was repeated three times with three biological replicates. The sequences and amplification efficiencies of primers used in this section are listed in Supplementary Table 13.

### Stress sensitivity assay

To assess the sensitivity of the transformants to different stress treatments, fresh 4.2 mm hyphal plugs from the wild-type P6497, CK, and transformants were transferred from 10% V8 plates to modified V8 medium plates containing 0.1 μg/mL pyraclostrobin, 0.5 μg/mL ametoctradin, 4 mM H$_2$O$_2$, 0.5 M sorbitol, or 0.5 M KCl, respectively. The plates were then cultured at 28 °C for 7 d in the dark. In addition, mitochondrial respiration inhibitors pyraclostrobin, ametoctradin, and mock treatment were supplemented with 100 mg/L salicylhydroxamic acid (SHAM) to inhibit bypass oxidation[65]. The diameter of each strain was measured, and the growth rate and inhibition rate were calculated. The inhibition rate was calculated as (CK−Growth rate on plates with treatment)/CK.

### *P. sojae* infection assays

To determine the virulence of *P. sojae*, mycelium was inoculated into soybean unifoliate leaves or the hypocotyls of etiolated seedlings, and the lesion areas and lengths were measured 48 h post-inoculation. To further evaluate differences in virulence, the relative fungal biomass in the inoculated hypocotyls at 48 hours post-inoculation was quantified by genomic DNA qPCR measurements of the ratio of pathogen to plant DNA. Amplification of the *PsACTB* from *P. sojae* and *GmCYP2* from *G. max* was performed using 50 ng of total genomic DNA from the infected material[66]; the quantitative primers used are given in Supplementary Table 13.

### Confocal microscopy

The transformants expressing fluorescently labeled target proteins were cultured in V8 liquid medium for 48 h or inoculated into etiolated seedlings for 2 h (during the infection stage) and observed using LSM 900 laser scanning microscope (Carl Zeiss, Germany) at specific excitation and emission wavelengths (excitation wavelengths: GFP, 488 nm; mCherry, 561 nm; and BFP, 610 nm).

### Quantification analysis of confocal images

For the analysis of colocalized puncta and mitochondrial acidification in *P. sojae* mycelia, we used over 20 puncta for quantification analysis with at least 3 biological replicates. The distribution of punctate localization of different colors was then calculated over the total number of punctate structures. When counting the punctate fluorescence composition of mCherry-GFP-PsAF5 and BFP-PsATG8, the blue puncta observed in the fluorescent photographs were categorized as "Only PsATG8", the yellow puncta as "Only PsAF5, unacidified", the red puncta in which the GFP was bleached as "acidic structure (ATG8 dissociation)", purple puncta in which the red and the blue co-localized as "Acidic structure", and puncta showing co-localization of all three colors fluorescence as "Unacidified colocalization".

### Protein extraction and co-immunoprecipitation

Transformants were cultured in liquid V8 medium for 3 d then total proteins were extracted using the BestBio Thick-wall microbial protein extraction kit. N-ethyl maleimide (NEM) was added when detecting PsATG8[22]. The resulting protein extract was incubated with 25 μL (0.25 mg) anti-Flag, anti-HA, or anti-Myc agarose beads (Thermo Fisher Scientific) for 3 h at 4 °C. The combined agarose beads were then collected by centrifugation and washed three times with pre-cooled wash buffer (50 mM Tris.HCl, 0.15 M NaCl, pH = 7.4). The immunoprecipitates were separated by SDS-PAGE and detected using the corresponding antibodies.

### Immunoprecipitation-mass spectrometry analysis

*PsAF5* was tagged with *3×Flag* and transferred into the *PsAF5* deletion mutant, then the resulting transformant was used for protein extraction as described above. The extracted proteins were used for capture of Flag-PsAF5 interacting proteins using the same steps as in the co-immunoprecipitation section.

The lyophilized peptide fractions were re-suspended in ddH$_2$O containing 0.1% formic acid, and 8 μL aliquots of the peptide suspensions were loaded into a nanoViper C18 trap column (Acclaim PepMap 100, 75 μm × 2 cm) trap column. The flow rate was 0.3 μL/min. The online chromatography separation was performed on the UltiMate 3000 RSLC nano (ThermoFisher). The trapping and desalting procedures were carried out with 20 μL of 100% solvent A (0.1% formic acid). Then, an elution gradient of 5–40% solvent B (80% acetonitrile, 0.1% formic acid) over 60 min was used on an analytical column (Acclaim PepMap RSLC, 75 μm × 25 cm C18-2 μm 100 Å). DDA (data-dependent acquisition) mass spectrum techniques were used to acquire tandem MS data on a ThermoFisher Q Exactive plus mass spectrometer (ThermoFisher) fitted with a Nano Flex ion source. Data were acquired using an ion spray voltage of 1.9 kV, and an interface heater temperature of 320 °C. For a full mass spectrometry survey scans, the target value was 3e$^6$ and the scan ranged from 350 to 1500 m/z at a resolution of 70,000 and a maximum injection time of 100 ms. For the MS2 scan, only spectra with a charge state of 2–5 were selected for fragmentation by higher-energy collision dissociation with a normalized collision energy of 28. The MS2 spectra were acquired in the ion trap in rapid mode with an AGC target of 1e$^5$ and a maximum injection time of 50 ms. Dynamic exclusion was set for 25 s.

The MS/MS data were analyzed for protein identification and quantification using Thermo Proteome Discoverer 2.5 software. The local false discovery rate per peptide was 1.0% after searching against the *P. sojae* Physo3_GeneCatalog database with a maximum of two missed cleavages.

## MBP pull-down assays

To perform MBP pull-down assays, we first produced constructs encoding MBP-tagged PsPHB2 protein, His-tagged PsATG8 protein, and His-tagged PsAF5 proteins using recombinant DNA techniques and then expressed them in *E. coli*. After harvesting the *E. coli* cells, the cells were disrupted using lysis buffer appropriate for the respective tag. The MBP-PsPHB2 and MBP-PsAF5 proteins were then incubated with Dextrin 6FF agarose beads and the supernatant was discarded. His-PsATG8 or PsAF5-His were subsequently purified using Ni-NTA resin and co-incubated with the bead-bound MBP proteins at 4 °C for 3 h. The resulting mixture was then subjected to pull-down centrifugation, and the co-precipitation of the His-tagged proteins was detected using an anti-His antibody.

## Isolation of mitochondria by sucrose gradient centrifugation

To separate and analyze mitochondria, we utilized sucrose gradient centrifugation. First, hyphae cultured in liquid for 3 d were homogenized using a dounce glass homogenizer with homogenization buffer (10 mM Tris/HCl, pH = 7.4, 0.6 M Sorbitol, 1 mM EDTA, 0.2% (W/V) BSA). Unbroken cells, nuclei, and large debris were then removed by successive centrifugation steps (1500 g and 3000 g). The resulting supernatant was centrifuged at 12,000 g, and the resulting pellet was resuspended in SEM buffer (10 mM MOPS/KOH, pH = 7.2, 250 mM Sucrose, 1 mM EDTA). A sucrose gradient was prepared in an ultra-clear centrifuge tube, and the crude mitochondria resuspended in SEM buffer were added to the top layer of the gradient. The tube was then centrifuged at 1,34,000 g, and intact mitochondria formed a brown band at the 60–32% sucrose interface[66]. Janus Green B can only stain intact, oxidatively active mitochondria[67]. A 1:1 mix of 0.2% Janus Green B (Merck) was used with freshly extracted mitochondria, followed by observation of mitochondrial activity using a light microscope. The freshly extracted mitochondria were initially fixed with a 2.5% glutaraldehyde solution. Subsequently, the treated mitochondria underwent post-fixation using 1% osmium tetroxide, dehydration in a series of acetone solutions, and the dehydrated material is infiltrated with Epox 812 prior to embedding. For visualization purposes, semithin sections were stained with methylene blue while ultrathin sections were cut using a diamond knife and subsequently stained with uranyl acetate and lead citrate. Finally, the sections were examined utilizing the JEM-1400-FLASH Transmission Electron Microscope (TEM).

## Immunoblot analysis

To detect the presence of specific proteins, we utilized immunoblotting. Proteins were fractionated using 10% SDS-PAGE gels (the formation of ATG8–PE was analyzed by 6 M urea SDS–PAGE, which has already been indicated in where it is used) and transferred to a PVDF membrane. The membrane was blocked using TBST buffer containing 10% skim milk powder and incubated overnight at 4 °C with the corresponding primary antibody. The membrane was then washed and incubated with a secondary antibody. The signals were amplified by HRP-streptavidin and detected by chemiluminescence assays. Antibodies used were: anti-Flag (1:8000; AB0008; Abways), anti-GFP (1:5000; AB0005; Abways), anti-MBP (1:5000; AB0029; Abways) antibody, anti-ACTB (1:8000; AB2001; Abways), anti-ATG8 (1:6000; AB77003; Abways), anti-ubiquitin (1:2000; AB40290; Abbkine) or anti-Myc (1:5000; PTM-5390; PTM BIO).

## Mitochondrial protein and DNA quantification

The relative levels of mitochondrial proteins and DNA are commonly employed to assess the number of mitochondria and monitor mitophagy[25,68]. In this study, mycelia were exposed to $H_2O_2$ (1 mM), pyraclostrobin (2 μg/mL), or ametoctradin (4 μg/mL) for 36 h, with or without Baf A1 (20 nM) treatment, while the solvent served as a control. Levels of mitochondrial proteins in appropriately tagged transformants were quantitated by immunoblotting with His-tag antibodies against AIF B (a mitochondrial matrix protein), mitochondrial carrier (MC) (an inner mitochondrial membrane protein), or anti-Flag antibodies against Cytc (a mitochondrial matrix protein), and comparison with cytoplasmic actin. For DNA quantitation, genomic DNA was then extracted using the CTAB method as previously described[62]. The extracted DNA was diluted to 100 ng/μL using NanoDrop (Thermo). The mitochondrial gene content was quantified using the qPCR method targeting the mitochondrial genes *ATPFO1* and *Cytb* and the nuclear gene *ACTB* using the primers listed in Supplementary Tables 13 and 14.

## Statistical and reproducibility

Statistical analysis was performed using Prism 9 (GraphPad, San Diego, CA, USA) and one-way ANOVA and Dunnett's multiple comparisons test. Pairwise comparisons were performed using Student's *t*-test. Data are reported as the mean ± SD or ±SEM, which is specified in the figure legend.

## Reporting summary

Further information on research design is available in the Nature Portfolio Reporting Summary linked to this article.

## Data availability

The information about gene involved in this article can be found in Supplementary Table 15. The mass spectrometry proteomics data generated in this study have been deposited in the ProteomeXchange with the identifier PXD044242. The authors declare that all data supporting the findings of this study are available within the article and its Supplementary Information files. Source data are provided with this paper.

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

## Acknowledgements
We are very grateful to Prof. Brett M. Tyler, Prof. Jinrong Xu (Purdue University), Prof. Cong Jiang (Northwest A&F University), and Prof. Liying Sun (Northwest A&F University) for their valuable suggestions. We thank Dr. Hua Zhao and Dr. Fengping Yuan (State Key Laboratory for Crop Stress Resistance and High-Efficiency Production, Northwest A&F University, Yangling, China) for confocal experimental assistance. Q.P. was supported by the National Natural Science Foundation of China (32102262), and X.L. was supported by the Innovation Capability Support Plan of Shaanxi Province of China (2020TD-035).

## Author contributions
W.L., X.L. and J.M. designed the study. X.L., J.M. and Q.P. supervised the study. W.L., H.Z., J.C. and B.R. performed the majority of experiments and data analyses. W.L., X.L. and J.M. wrote the manuscript. All authors read the manuscript.

## Competing interests
The authors declare no competing interests.
