## [Peer Review File · Nature Communications]

REVIEWER COMMENTS

Reviewer #1 (Remarks to the Author):

The manuscript by Li and colleagues investigates how *Phytophthora sojae* copes with extensive ROS production during infection. This is a remarkable study of oomycete infection cell biology that offers a completely new view of a field dominated by studies on effectors inhibiting host ROS burst. The manuscript has some minor English glitches, and some figures could be improved. However, the experiments are extremely well-designed and robust. The statistical analysis is excellent, and the authors have provided all the raw data. I am impressed by the work presented here and very enthusiastic about its impact on the field.

Minor remarks:

To improve figure readability, especially of the ATG8 channel, please use grayscale for the individual channels and colours for the merge only.

Add some arrowheads in the relevant part of the pictures to guide the reader.

Fig 3a could be moved to supplementary information.

Fig 6f contains a merged picture with a bright field background, while other pictures do not. Check consistency.

Fig S8 is unreadable. Text is often too small.

Minor editing comments:

Please check the manuscript for English.

26-27 'Unlike PHB2 can recruit ATG8 directly to mitochondria in animals, PsPHB2 cannot recruit': missing word?

30 'plant pathogenic': missing hyphen

31 'the IMM protein as a receptor to mediate mitophagy is conserved in eukaryotes': rephrase e.g. the mitophagy inducer IMM is conserved...

35 'the innate immune system of host plants': the plant innate immune system

66 'phagophore': explain the term

79 'plant pathogen fungus': fungal plant pathogen, or plant-pathogenic fungus

126 'the combination with a TPR domain': Tetratricopeptide repeat (TPR)

Reviewer #2 (Remarks to the Author):

The manuscript entitled “PsAF5 functions as an essential adapter for PsPHB2-mediated mitophagy under ROS stress in *Phytophthora sojae*” shows a novel strategy to induce mitochondrion-targeting autophagy termed mitophagy in a plant-pathogenic fungi, *Phytophthora sojae*. When the fungi are exposed to oxidative stress, the cytoplasmic protein PsAF5 acts as an adapter protein between autophagosomal membrane-anchored PsATG8 and mitochondrial inner-membrane protein PsPHB2 to induce mitophagy that eliminates ROS-accumulating mitochondria. This type of mitophagy contributes to ROS tolerance of the fungi, which is important in its infection to host plants. Overall, the study is well organized, and the data supports the main conclusion. However, some results look inconsistent. The following points are my concerns.

i) When the behavior of PsAF5 is monitored by GFP-mCherry tandem tag (Figure 2f), the puncta indicating the mCherry signal without GFP signal appear where BFP-PsATG8 also accumulates. The authors called such puncta “acidified” autophagic structures since GFP fluorescence quenches and only RFP (mCherry) signal accumulates when the tandem tag is transported into the acidic organelles. The authors considered the accumulation of such acidified autophagic structures as an indicator of autophagy activity during H₂O₂ or infection treatment. Such compartments probably correspond to autolysosomes in mammalian cells. However, in supplemental Figure 5a, condensed GFP-PsAF5 signal beautifully colocalizes with BFP-PsATG8 signal. No puncta exhibited only BFP-ATG8 signal that should correspond to acidic autophagic structures. Figure 2h shows around 70% of such puncta are acidic structures that exhibit mCherry signal from mCherry-GFP-PsAF5 and BFP signal from BFP-PsATG8. Would you consider explaining why such differences happen.

ii) I also point out that many puncta apparently contain mCherry and GFP signal during H₂O₂ and infection treatment (Figure 2f). However, figure 2g and 2h indicates that around 70% of such puncta show mCherry signal without GFP and colocalize with BFP-ATG8 signal in cells expressing unmutated mCherry-GFP-PsAF5 and BFP-PsATG8. There are huge differences between figure 2d and 2h results. I understood that the figure 2g and 2h experiments were performed with H₂O₂ treatment since the study found that such ROS stress induces the colocalization between PsAF5 and PsATG8. Is this my misunderstanding? The main text and legend do not explain this point. Furthermore, the method section does not explain how to process confocal images for quantitative analysis.

iii) Although the authors concluded as “PsAF5 interacts with ATG8 via AIM1 motif”, the current data do not directly evaluate the interaction between mutated PsAF5 and PsATG8. Such experiments are required for the conclusion and is not so difficult since the authors already established the in vitro and in vivo pull down assay for the evaluation of wild-type PsAF5 and PsATG8 interaction (Figure 2c and 2d). I was also wonder why the condensation of GFP signal of mCherry-GFP-PsAF5 did not occur in cells expressing PsAF5 harboring MuAIM2–4 mutations even though these mutated PsAF5 do not affect its interaction with PsATG8 and the phenotypes during H₂O₂ treatment and infection efficiency. Additionally, Figure 5c indicates that PsAF5 without ANK domain (deltaANK) abolishes mitochondrial accumulation of PsATG8 and PsAF5 similarly with PsAF5 harboring MuAIM1 mutation. However,

mCherry-GFP PsAF5-deltaANK did not affect its colocalization with BFP-ATG8 in Figure 2g. Please consider explain the reason why.

iv) In the current data, mitophagy flux is only evaluated by the observations of mitochondria or PsAF5 labeled by mCherry-GFP tandem tag. The study does not show the validity of such assay. I think the authors should check the appearance of mCherry-positive and GFP-negative puncta (in cells expressing mCherry-GFP-tagged mitochondria) do not appear in the mutants of autophagy to confirm such observations certainly indicate the occurrence of mitophagy.

Reviewer #3 (Remarks to the Author):

This manuscript reported an interplay between autophagy(mitophagy) and oxidative stress of *P. sojae* and its host. Although a lot of work has been done by the authors, the interpretation of the results is constrained by technical limitations.

The authors used Immunoprecipitation-Mass Spectrometry (IP-MS) methods to identify PsAF5 as an adapter protein of IMM mitophagy receptor PHB2, which serves as a bridge for PHB2 to recruit ATG8. The authors failed to provide the details of this approach in the paper. They provided supporting evidence but not the major MS/MS experiment on which they claimed their finding.

The authors used a mutant of PsAF5 (silenced/non-functional gene). They should also include an overexpression mutant in this study to get a complete understanding of the adapter role of PsAF5 and its recruitment of ATG8

Difficult to understand the figures without detailed figure captions. It is recommended that all figures caption be self-explanatory. In all microscopy images make sure the size of the scale bar is the same between mock and treatments. Different size scale bars with the same length are misleading.

Fig1 (e, f, g) – how many days post infections (DPI)? or is it diphenyleneiodonium (DPI) inhibition tests? It is misleading when not clear.

Fig2 (e) - PH to pH

Fig2 (g) Label for the last image in (g) is missing. The control condition is missing. The scale bar looks different in each image, does it measure the same?

Fig3 (e): Janus green B staining procedure missing from material and methods. The image is too small to see any results.

Fig3 (i,j) - 3 mM H₂O₂ sensitivity and pathogenicity of different AIM mutants of PsAF5. In Fig 1&2 H₂O₂ was 1mM. Why did you decide to use a different concentration for this experiment?

Line 207: The results were derived from three (n=18) and two (n=12) biological replicates. For which experiment which replicate was used?

Fig7: PE stands for what? Lipidated? Ub? - please include in the figure description, also include full forms of BFP- Blue fluorescent protein,

Some sections of the methods and materials need to include more details. For example Line 566: In the qPCR analysis section- Did you follow MIQE guidelines while performing qPCR? Was the primer efficiency of your primers tested? The housekeeping genes exhibit variable expression levels depending on the organism, its developmental stage, its response to external stimuli, and the experimental conditions. Actin B (ACTB) is used as a reference gene and can fluctuate upon the treatment with pharmacological agents or under some physiological and pathological conditions. Please provide a reference for actin (ACTB) being a suitable housekeeping gene for this experiment. The primer details are not in supplementary Table 1.

Line 581: Was the stress test performed in light or dark?

Line 591: In *P. sojae* infection assays - why was the β -tubulin gene used and actin used for amplification from DNA? I don't see any results related to β -tubulin in the paper.

line 598- How much protein and antibody was used? How did you confirm the protein concentration?

line 605: provide more mass spectrometry details. Make a separate paragraph for the MS/MS method. These experiment details are important for the paper.

Line 612: What do you mean by flow fluid?

line 626: Sucrose gradient centrifugation: 134000 g? Mitochondria will break at that high "g". How are you making sure you are taking the exact same amount of mitochondria for each experiment after sucrose gradient extraction? Have you performed any viable cell assays? How are you sure the mitochondria you are using in your experiment are alive?

line 645: The qPCR method with the primers is not listed in Supplementary Table 1.

There are several grammatical mistakes all over the paper for example :

line 223: puncta?

Line 628: Immunoblotting blotting- the word blotting is written twice

Line 276: H₂O₂ treatment can promote the transfer of PsAF5 from cytoplasm to mitochondria from cytoplasm. The word from cytoplasm is repetitive.

Supplemental table (2-7) information missing from the main paper. I see only the supplemental table1 too is cross-referenced incorrectly

The authors should make sure these mistakes are addressed properly throughout the manuscript.

Please check the references as I found several references for eg no. 6,12,18,23 etc. have only one author name and et al., at the end; whereas other references have more than 1 author name. The authors should follow the citation style recommended by the journal and make sure it is consistent for all the references.

Response Letter to Nature Communications Submission

Reviewer #1:

Reviewer's Comment 1.0: The manuscript by Li and colleagues investigates how *Phytophthora sojae* copes with extensive ROS production during infection. This is a remarkable study of oomycete infection cell biology that offers a completely new view of a field dominated by studies on effectors inhibiting host ROS burst. The manuscript has some minor English glitches, and some figures could be improved. However, the experiments are extremely well-designed and robust. The statistical analysis is excellent, and the authors have provided all the raw data. I am impressed by the work presented here and very enthusiastic about its impact on the field.

Author's Response 1.0: We thank the editor for your kind comments and responded in a point-by-point manner.

Reviewer's Comment 1.1: To improve figure readability, especially of the ATG8 channel, please use grayscale for the individual channels and colours for the merge only.

Author's Author's Response 1.1: The relevant figures have been modified as you suggested: BFP-ATG8 channels use grayscale, merge uses colors.

Reviewer's Comment 1.2: Add some arrowheads in the relevant part of the pictures to guide the reader.

Author's Response 1.2: In accordance with your request, triangle arrows have been added to the positions that need to be highlighted.

Reviewer's Comment 1.3: Fig 3a could be moved to supplementary information.

Author's Response 1.3: We accept this suggestion. **Fig. 3a** has now been moved to become **Supplementary Table 8**.

Reviewer's Comment 1.4: Fig 6f contains a merged picture with a bright field background, while other pictures do not. Check consistency.

Author's Response 1.4: The bright field background in **Fig. 7f** has been removed, according to your suggestion.

Reviewer's Comment 1.5: Fig S8 is unreadable. Text is often too small.

Author's Response 1.5: The text in **Supplementary Fig. 9 (Previously Fig S8)** has been enlarged to size 6, which is consistent with the other figures and meets the font size requirements of the journal.

Reviewer's Comment 1.6: 26-27 'Unlike PHB2 can recruit ATG8 directly to mitochondria in animals, PsPHB2 cannot recruit': missing word?

Author's Response 1.6: The first half of the sentence refers to PHB2 and ATG8 in animals, while the second half of the sentence refers to PHB2 and ATG8 in *Phytophthora sojae*. It has been revised in text **Line 22-24: "Unlike animal PHB2 that can recruit ATG8 directly to mitochondria, PsPHB2 in *P. sojae* cannot recruit PsATG8 to stressed mitochondria without PsAF5"**.

Reviewer's Comment 1.7: 30 'plant pathogenic': missing hyphen

Author's Response 1.7: Corrected as suggested: **Line 27, "plant-pathogenic"**.

Reviewer's Comment 1.8: 31 ' the IMM protein as a receptor to mediate mitophagy is conserved in eukaryotes': rephrase e.g. the mitophagy inducer IMM is conserved...

Author's Response 1.8: Modified as follows: **Line 27-29 "that mitophagy induction by IMM proteins is conserved in eukaryotes, but with differences in the details of ATG8 recruitment"**.

Reviewer's Comment 1.9: 35 'the innate immune system of host plants': the plant innate immune system

Author's Response 1.9: Modified as suggested: **Line 32 "the plant innate immune system in response to..."**.

Reviewer's Comment 1.10: 66 'phagophore': explain the term

Author's Response 1.10: Modified as follows: **Line 68-70 "The recruited ATG8 plays a central role in assembly of the phagophore (autophagosome precursor), in cargo recruitment, and in autophagosome membrane bilayer maturation ..."**.

Reviewer's Comment 1.11: 79'plant pathogen fungus': fungal plant pathogen, or plant-pathogenic fungus

Author's Response 1.11: Modified as follows: **Line 83 "plant-pathogenic fungus"**.

Reviewer's Comment 1.12: 126 'the combination with a TPR domain': Tetratricopeptide repeat (TPR)

Author's Response 1.12: Modified in the revised text: **Line 137 "the combination with a tetratricopeptide repeat (TPR) domain"**.

Reviewer #2:

Reviewer's Comment 2.0: The manuscript entitled "PsAF5 functions as an essential adapter for PsPHB2-mediated mitophagy under ROS stress in *Phytophthora sojae*" shows a novel strategy to induce mitochondrion-targeting autophagy termed mitophagy in a plant-

pathogenic fungi, *Phytophthora sojae*. When the fungi are exposed to oxidative stress, the cytoplasmic protein PsAF5 acts as an adapter protein between autophagosomal membrane-anchored PsATG8 and mitochondrial inner-membrane protein PsPHB2 to induce mitophagy that eliminates ROS-accumulating mitochondria. This type of mitophagy contributes to ROS tolerance of the fungi, which is important in its infection to host plants. Overall, the study is well organized, and the data supports the main conclusion. However, some results look inconsistent. The following points are my concerns.

Author's Response 2.0: Thank you for your careful evaluation of this manuscript.

Reviewer's Comment 2.1: When the behavior of PsAF5 is monitored by GFP-mCherry tandem tag (Figure 2f), the puncta indicating the mCherry signal without GFP signal appear where BFP-PsATG8 also accumulates. The authors called such puncta "acidified" autophagic structures since GFP fluorescence quenches and only RFP (mCherry) signal accumulates when the tandem tag is transported into the acidic organelles. The authors considered the accumulation of such acidified autophagic structures as an indicator of autophagy activity during H₂O₂ or infection treatment. Such compartments probably correspond to autolysosomes in mammalian cells. However, in supplemental Figure 5a, condensed GFP-PsAF5 signal beautifully colocalizes with BFP-PsATG8 signal. No puncta exhibited only BFP-ATG8 signal that should correspond to acidic autophagic structures. Figure 2h shows around 70% of such puncta are acidic structures that exhibit mCherry signal from mCherry-GFP-PsAF5 and BFP signal from BFP-PsATG8. Would you consider explaining why such differences happen.

Author's Response 2.1: The localization of GFP-PsAF5 and BFP-PsATG8 in **Supplementary Fig. 5a** was observed within a short period of time after H₂O₂ treatment (1 mM, 1 h), which is the same as the treatment conditions in the second row of **Fig. 2f** (conditions, i.e., 1 mM, 1 h, are now stated in the legend); in these conditions the structures containing GFP tagged protein likely have not had time to acidify. However, the purpose of **Supplementary Fig. 5a** was to emphasize the co-localization of PsAF5 and PsATG8 that developed under those conditions, so the images primarily illustrate structures exhibiting co-localization, rather than structures that do not. In response to your comment, the fluorescence images have been replaced with images illustrating co-localization plus occasional non-co-localization (presumably due to acidification).

The results shown **Fig. 2g, 2h**, however, were obtained in the ABSENCE of H₂O₂ treatment or infection, i.e., comparable to the first row in **Fig. 2f**. So, there are more acidic structures that exhibit mCherry, but not GFP, signals from mCherry-GFP-PsAF5 and BFP signal from BFP-PsATG8 due to the accumulation of acidic structures generated by normal autophagy processes within the mycelium.

Reviewer's Comment 2.2: I also point out that many puncta apparently contain mCherry and GFP signal during H₂O₂ and infection treatment (Figure 2f). However, figure 2g and 2h indicates that around 70% of such puncta show mCherry signal without GFP and colocalize with BFP-ATG8 signal in cells expressing unmutated mCherry-GFP-PsAF5 and BFP-PsATG8. There are huge differences between figure 2d (f) and 2h results. I understood that the figure 2g and 2h experiments were performed with H₂O₂ treatment since the study found that such ROS stress induces the colocalization between PsAF5 and PsATG8. Is this my misunderstanding? The main text and legend do not explain this point. Furthermore, the method section does not explain how to process confocal images for quantitative analysis.

Author's Response 2.2: Indeed, the lack of description in the manuscript led to your misunderstanding; in fact, **no infection or H₂O₂ treatment was performed in Fig. 2g and 2h**, consistent with the Mock line in **Fig. 2f**; this information has now been added to the figure legends in **figure 2g**.

In **Fig. 2g**, the ability to form puncta that exhibit mCherry, but not GFP, signal and colocalize with BFP-ATG8 signal, in cells expressing unmutated mCherry-GFP-PsAF5 and BFP-PsATG8 could indicate the accumulation of "acidified structures" resulting from basal level autophagy in the hyphae. In **Fig. 2f**, after infection or H₂O₂ treatment, many puncta apparently contain mCherry and GFP signal, which are described "**This suggests the formation of newly generated autophagic structures that have not undergone acidification**" (Lines 272-273), and is also consistent with the formation of punctate GFP signals after H₂O₂ treatment in **Supplemental Fig. 5a**. Furthermore, the Method section now includes a new paragraph titled "**Quantification analysis of confocal images**" (Lines 740) which outlines the process for quantitatively analyzing confocal images.

Reviewer's Comment 2.3: Although the authors concluded as "PsAF5 interacts with ATG8 via AIM1 motif", the current data do not directly evaluate the interaction between mutated PsAF5 and PsATG8. Such experiments are required for the conclusion and is not so difficult since the authors already established the in vitro and in vivo pull-down assay for the evaluation of wild-type PsAF5 and PsATG8 interaction (Figure 2c and 2d).

Author's Response 2-3: Thanks for the suggestion, Co-IP and pull-down tests have been added to **Fig. 2i** and **2j**.

Reviewer's Comment 2.4: I was also wonder why the condensation of GFP signal of mCherry-GFP-PsAF5 did not occur in cells expressing PsAF5 harboring MuAIM2-4 mutations even though these mutated PsAF5 do not affect its interaction with PsATG8 and the phenotypes during H₂O₂ treatment and infection efficiency. Additionally, Figure 5c indicates that PsAF5 without ANK domain (deltaANK) abolishes mitochondrial accumulation of PsATG8 and PsAF5 similarly with PsAF5 harboring MuAIM1 mutation.

However, mCherry-GFP PsAF5-deltaANK did not affect its colocalization with BFP-ATG8 in Figure 2g. Please consider explain the reason why.

Author's Response 2.4: Indeed, the MuAIM2-4 mutation on PsAF5 does not affect its interactions with PsATG8 (Fig. 2i and 2j). Therefore, the localization of PsAF5 with the MuAIM2-4 mutation should be consistent with that of unmutated PsAF5, as shown in Fig. 2g. However, please note that the conditions of Fig. 2g, did not include treatment with H₂O₂. We apologize for that unclear information about Fig. 2g.

In response to the second part of your comment, Fig. 7e (original Fig. 5c) demonstrates that both the AIM1 mutation and truncation of the ANK structural domain result in reduced accumulation of PsATG8 and PsAF5 in the mitochondria. However, it is important to note that the truncated ANK structure of PsAF5 still co-localizes with PsATG8 (Fig. 2g). The newly added Pull-down and Co-IP experiments reveal that the truncation of the ANK domain does not affect the interactions between PsAF5 and PsATG8 (Figures. 2i and 2j), but only affect the interactions between PsAF5 and PsPHB2 (Fig. 6d, 6e and 6f). After truncation of the ANK domain, the puncta where PsAF5 and PsATG8 co-localize may not be in the mitochondria (Fig. 2g). Thus, our model is that the AIM1 motif mediates the binding of PsAF5 to ATG8 while the ankyrin domain mediates binding to prohibitin 2.

Reviewer's Comment 2.5: In the current data, mitophagy flux is only evaluated by the observations of mitochondria or PsAF5 labeled by mCherry-GFP tandem tag. The study does not show the validity of such assay. I think the authors should check the appearance of mCherry-positive and GFP-negative puncta (in cells expressing mCherry-GFP-tagged mitochondria) do not appear in the mutants of autophagy to confirm such observations certainly indicate the occurrence of mitophagy.

Author's Response 2.5: ATG7 has been verified to play an important role in ATG8 lipidation and autophagy activation in both mammals and fungi¹. We obtained a new *PsATG7* homozygous knockout mutant (Supplementary Fig. 8a) and performed a series of experiments with it. In the *PsATG7* knockout mutant, the activation of autophagy in ROS stress is blocked (Fig. 5b).

The appearance of mCherry-positive and GFP-negative puncta (in cells expressing mCherry-GFP-tagged mitochondria) does not appear in the $\Delta PsATG7$ mutants (Fig. 5c), and the autophagy inhibitor bafilomycin A1 is also able to inhibit this process (Fig. 5d).

Reviewer #3:

Reviewer's Comment 3.0: This manuscript reported an interplay between autophagy(mitophagy) and oxidative stress of *P. sojae* and its host. Although a lot of work has been done by the authors, the interpretation of the results is constrained by technical limitations.

Author's Response 3.0: We thank the reviewer for your thoughtful comments and we have responded in a point-by-point manner.

Reviewer's Comment 3.1: The authors used Immunoprecipitation-Mass Spectrometry (IP-MS) methods to identify PsAF5 as an adapter protein of IMM mitophagy receptor PHB2, which serves as a bridge for PHB2 to recruit ATG8. The authors failed to provide the details of this approach in the paper. They provided supporting evidence but not the major MS/MS experiment on which they claimed their finding.

Author's Response 3.1: Thanks to your advice, the missing part of the Methods has been added in the **Materials and Methods** section of the manuscript (**Line 762**) under the title "Immunoprecipitation-mass spectrometry analysis", and the protein mass spectrometry raw data has been uploaded to the ProteomeXchange database (**Line 861**) as requested by the editor.

Reviewer's Comment 3.2: The authors used a mutant of *PsAF5* (silenced/non-functional gene). They should also include an overexpression mutant in this study to get a complete understanding of the adapter role of *PsAF5* and its recruitment of **ATG8**.

Author's Response 3.2: An 8-fold overexpression of *PsAF5* strain was obtained after following your suggestion (**Supplementary Fig. 8b**), and a significantly higher proportion of activation of ATG8 on mitochondria was detected in the overexpressed strain than in the wild-type (**Supplementary Fig. 8c and 8d**). In *PsAF5* overexpressing strains, the interaction between PsPHB2 and activated PsATG8 was also enhanced under ROS stress (**Supplementary Fig. 8e**).

Reviewer's Comment 3.3: Difficult to understand the figures without detailed figure captions. It is recommended that all figures caption be self-explanatory. In all microscopy images make sure the size of the scale bar is the same between mock and treatments. Different size scale bars with the same length are misleading.

Author's Response 3.3: All figure captions in the main text have been carefully revised to ensure full details and descriptions are present. The scale bars of the microscopy images within each panel have been all adjusted to be consistent by scaling of the images.

Reviewer's Comment 3.4: Fig1 (e,f, g) – how many days post infections (DPI)? or is it diphenyleneiodonium (DPI) inhibition tests? It is misleading when not clear.

Author's Response 3.4: To avoid misunderstandings, the full name for the abbreviation DPI (diphenyleneiodonium) has been noted in the caption for **Fig. 1e-g (Line 188)**.

Reviewer's Comment 3.5: Fig2 (e) - PH to pH

Author's Response 3.5: Corrected in **Fig. 2e**.

Reviewer's Comment 3.6: Fig2 (g) label for the last image in (g) is missing. The control condition is missing. The scale bar looks different in each image, does it measure the same?

Author's Response 3.6: Label for the last image in **Fig. 2g** shows the subcellular localization of control fluorescence expressing only the mCherry-GFP tandem label, to avoid confusion the "-" has been changed to "**Free tag**".

The fluorescent hyphal material in **Fig. 2g** was not infected or treated with H₂O₂ (this has been added to the article in **Line 245 "without any H₂O₂ or infection treatment"**), so no control condition was set in the article. The scale bars in **Fig. 2g** have been adjusted to be consistent by image scaling.

Reviewer's Comment 3.7: Fig3 (e): Janus green B staining procedure missing from material and methods. The image is too small to see any results.

Author's Response 3.7: The staining method for Janus green B has been added to the section "**Isolation of mitochondria by sucrose gradient centrifugation**" in the materials and methods (**Line 800**). Probably because the mitochondria of *Phytophthora sojae* are too small, we did not get better mitochondrial pictures with a light microscope (BX53F, Olympus) than before, but the rectangular framed area of the picture was enlarged as much as possible. In order to see more details of the mitochondria, we simultaneously observed the isolated mitochondria using transmission electron microscopy, and the obtained photos were provided at the same time as the photos taken by the light microscope (**Fig. 3e**).

Reviewer's Comment 3.8: Fig3 (i,j) - 3 mM H₂O₂ sensitivity and pathogenicity of different AIM mutants of PsAF5. In Fig 1&2 H₂O₂ was 1mM. Why did you decide to use a different concentration for this experiment?

Author's Response 3.8: In order to obtain a more clearly observe any phenotypes in the AIM mutants, a higher treatment concentration of H₂O₂ was originally used. According to your advice to improve consistency, we have replaced the data in **Fig 2i** with results from 1 mM H₂O₂ and the updated **Fig 2i** together with **2j** have been moved to the supplement to become **Supplementary Fig. 6a** and **6b**.

Reviewer's Comment 3.9: Line 207: The results were derived from three (n=18) and two (n=12) biological replicates. For which experiment which replicate was used?

Author's Response 3.9: Based on your suggestion, the experiment in **Supplementary Fig. 6a** (original **Fig 2i**) was repeated with a new 1 mM H₂O₂ treatment, where the number of biological replicates was adjusted to match the **Supplementary Figures 6b** (original **Fig 2j**) to avoid confusion (**Line 1123**).

Reviewer's Comment 3.10: Fig7: PE stands for what? Lipidated? Ub? - please include in the figure description, also include full forms of BFP- Blue fluorescent protein,

Author's Response 3.10: PE stands for "lipidated", meaning PsATG8 conjugated to phosphatidylethanolamine (PE); this has been added to the **Fig. 8** description (**Line 529**), as has an explanation of "Ub" (**Line 533**). Furthermore, where abbreviations such as BFP (**Line 241**) appeared for the first time in the text, the full name also has been added.

Reviewer's Comment 3.11: Some sections of the methods and materials need to include more details. For example, Line 566: In the qPCR analysis section- Did you follow MIQE guidelines while performing qPCR? Was the primer efficiency of your primers tested? The housekeeping genes exhibit variable expression levels depending on the organism, its developmental stage, its response to external stimuli, and the experimental conditions. Actin B (ACTB) is used as a reference gene and can fluctuate upon the treatment with pharmacological agents or under some physiological and pathological conditions. Please provide a reference for actin (ACTB) being a suitable housekeeping gene for this experiment. The primer details are not in supplementary Table 1.

Author's Response 3.11: qPCR was performed in accordance with the MIQE guidelines. The amplification efficiencies of the primers were tested and documented in the **Supplementary Table 13**. *PsActin* as a suitable housekeeping gene is a reference from the literature^{2,3,4}, and the cited references have been added to the main text. Many articles^{e.g.2,3,4}, used actin as the reference gene in qPCR analysis of different developmental stages of *Phytophthora sojae*, so actin was selected as the reference gene in this experiment.

The order of the **Supplementary Tables** has been rearranged and the order in which they appear in the text has been adjusted. The sequences of the primers for qPCR were transferred to **Supplementary Table 13**.

Reviewer's Comment 3.12: Line 581: Was the stress test performed in light or dark?

Author's Response 3.12: The stress test was performed in the **dark**, and the description has been updated in this section in **Line 717**.

Reviewer's Comment 3.13: In *P. sojae* infection assays - why was the β -tubulin gene used and actin used for amplification from DNA? I don't see any results related to β -tubulin in the paper.

Author's Response 3.13: In **Line 729** contained a mistake. Although both the *actin* and β -*tubulin* genes are commonly used for genomic DNA qPCR in *P. sojae*, our study utilized only *PsACTB* for *P. sojae* and *GmCYP2* for *G.max* for genomic DNA qPCR measurements of pathogen biomass. Note that because these assays utilize genomic DNA as substrate, the question of stable expression does not arise. Appropriate references have been cited in the main text. The primer sequences used are given in **Supplementary Table 13** and information about all genes is are all noted in **Supplementary Table 15**.

Reviewer's Comment 3.14: line 598- How much protein and antibody were used? How did you confirm the protein concentration?

Author's Response 3.14: In the "**Protein extraction and co-immunoprecipitation**" section (Line 752), the expression levels of proteins with different tags were identified by western blotting after protein extraction. Protein samples with consistent expression of tagged proteins were selected for Co-IP experiments. The dosage of antibody beads used was according to the company's instructions, and the dosage (0.25 mg) has been added to that Methods section (Line 756).

After co-immunoprecipitation, coomassie brilliant blue was used to preliminarily adjust the loading volume to be consistent before protein loading, and the sample loadings were adjusted for the second time according to the WB results of the directly precipitated protein by the antibody. For example, the post-IP protein in **Fig. 6b** was loaded according to the amount of protein indicated by the Flag-PsAF5 WB band.

Reviewer's Comment 3.15: line 605: provide more mass spectrometry details. Make a separate paragraph for the MS/MS method. These experiment details are important for the paper.

Author's Response 3.15: The MS/MS methods have been supplemented in detail and placed in a separate section with the title "**Immunoprecipitation-mass spectrometry analysis**" on line 762.

Reviewer's Comment 3.16: Line 612: What do you mean by flow fluid?

Author's Response 3.16: "flow fluid" means "**supernatant**", which has been modified in Line 795 the original text.

Reviewer's Comment 3.17: line 626: Sucrose gradient centrifugation: 134000 g? Mitochondria will break at that high "g". How are you making sure you are taking the exact same amount of mitochondria for each experiment after sucrose gradient extraction? Have you performed any viable cell assays? How are you sure the mitochondria you are using in your experiment are alive?

Author's Response 3.17: When extracting the mitochondria, centrifugation was performed using a centrifugal force of 134,000 g as described in the references⁵. This paper is widely cited, and the authors state that "**This method enables the isolation of highly pure mitochondria that are essentially free of contamination by other organelles and retain their structural and functional integrity after their purification.**"

At the beginning of the extraction of mitochondria from different samples, samples of hyphae with the same fresh weight taken after weighing. The number of mitochondria in each sample is further adjusted according to the WB results with the inner mitochondrial membrane marker protein (PHB2) (**Fig 3g; 6g; 7c, d**).

Janus Green B can only stain intact, oxidatively active mitochondria, and thus was used to confirm that the isolated mitochondria are intact and alive. Furthermore, the isolated mitochondria were observed by transmission electron microscopy (TEM) to further demonstrate the integrity of the extracted mitochondria (Fig 3f).

Reviewer's Comment 3.18: line 645: The qPCR method with the primers is not listed in Supplementary Table 1.

Author's Response 3.18: The order of the **Supplementary Tables** has been rearranged and the order in which they appear in the text has been adjusted. The qPCR method is described in the Methods section (**Line 700**), as before, and the primers are now listed in **Supplementary Table 13**.

There are several grammatical mistakes all over the paper for example:

Reviewer's Comment 3.19: line 223: puncta?

Author's Response 3.19: We apologize if our use of the term "puncta" was not clear. According to the Merriam-Webster Medical Dictionary a punctum (plural: puncta; adjective punctate) is a small area marked off from a surrounding surface. We have used this term (as have many others) to indicate small, compact region of prominent fluorescence that indicates the presence of a sub-cellular structure, such as an organelle or macromolecular complex. In order to improve clarity, we now refer "**punctate co-localization**" in **Line 270**.

Reviewer's Comment 3.20: Line 628: Immunoblotting blotting- the word blotting is written twice

Author's Response 3.20: It has been corrected in the revised text: "**Immunoblot analysis**" in **Line 826**.

Reviewer's Comment 3.21: Line 276: H₂O₂ treatment can promote the transfer of PsAF5 from cytoplasm to mitochondria from cytoplasm. The word from cytoplasm is repetitive.

Author's Response 3.21: The first "from cytoplasm" has been deleted from the text **Line 336**.

Reviewer's Comment 3.22: Supplemental table (2-7) information missing from the main paper. I see only the supplemental table1 too is cross-referenced incorrectly.

Author's Response 3.22: The order of the **Supplementary Tables** has been rearranged and the order in which they appear in the text has been adjusted. The **supplemental tables (1-15)** are now referenced directly in the relevant positions in the text.

Reviewer's Comment 3.23: Please check the references as I found several references for eg no. 6,12,18,23 etc. have only one author name and et al., at the end; whereas

other references have more than 1 author name. The authors should follow the citation style recommended by the journal and make sure it is consistent for all the references.

Author's Response 3.23: We rechecked the NC's citation requirements as well as several recent articles, and confirmed that all the authors must be listed when there are 6 or less than, **but for more than 6 authors, only the first author is named, and the others indicated et al.**

References mentioned in the Response to Reviewers

1. Mizushima, N., Yoshimori, T. & Ohsumi, Y. The role of atg proteins in autophagosome formation. *Annu Rev Cell Dev Biol* **27**, 107–132 (2011).
2. Wang, L. *et al.* Effector gene silencing mediated by histone methylation underpins host adaptation in an oomycete plant pathogen. *Nucleic Acids Res* **48**, 1790–1799 (2020).
3. Qiu, X. *et al.* The Phytophthora sojae nuclear effector PsAvh110 targets a host transcriptional complex to modulate plant immunity. *Plant Cell* **35**, 574–597 (2023).
4. Ma, Z. *et al.* A paralogous decoy protects Phytophthora sojae apoplastic effector PsXEG1 from a host inhibitor. *Science (1979)* **355**, 710–714 (2017).
5. Gregg, C., Kyryakov, P. & Titorenko, V. I. Purification of mitochondria from yeast cells. *J Vis Exp* 15–16 (2009) doi:10.3791/1417.

REVIEWERS' COMMENTS

Reviewer #1 (Remarks to the Author):

The authors successfully addressed all my concerns.

Reviewer #2 (Remarks to the Author):

The additional experiments and explanations improved the manuscript and my concerns were almost resolved. Please consider adding the top labels of the images in the Figure 5c and 5d.

Reviewer #3 (Remarks to the Author):

The authors response to my comments is acceptable.

For someone else to repeat the mass spec experiment please include your tune file setting and flow rate.
For LC please write the gradient in detail.

756- Informationfiles- include space Information files

Response Letter to Nature Communications Submission

Reviewer #1:

Reviewer's Comment 1.0: The authors successfully addressed all my concerns.

Author's Response 1.0: Thank you for your careful evaluation of this manuscript.

Reviewer #2:

Reviewer's Comment 2.0: The additional experiments and explanations improved the manuscript and my concerns were almost resolved.

Author's Response 2.0: Thank you for your careful evaluation of this manuscript.

Reviewer's Comment 2.1: Please consider adding the top labels of the images in the Figure 5c and 5d.

Author's Response 2.1: Thanks for the suggestion, top labels have been added to the images in the Fig. 5c and 5d.

Reviewer #3:

Reviewer's Comment 3.0: The authors response to my comments is acceptable.

Author's Response 3.0: We thank the reviewer for your thoughtful comments.

Reviewer's Comment 3.1: For someone else to repeat the mass spec experiment please include your tune file setting and flow rate. For LC please write the gradient in detail.

Author's Response 3.1: Due to excessively long description of "Immunoprecipitation-mass spectrometry analysis" section, the "tune file setting", "flow rate", and "gradient in detail" have been submitted to ProteomeXchange database under accession code PXD044242. In the revision, we incorporated the flow rate into our mass spectrometry experiment. Additional detailed information will be accessible on the website (<http://proteomecentral.proteomexchange.org/cgi/GetDataset?ID=PXD044242>) subsequent to publication of this article.

Reviewer's Comment 3.2: 756- Informationfiles- include space Information files.

Author's Response 3.2: It has been corrected in the revised text: "Information files" in Line 653.